# CANDI: Hybrid Discrete-Continuous Diffusion Models

Patrick Pynadath [1]   Jiaxin Shi [2]   Ruqi Zhang [1]

## Abstract

While continuous diffusion has shown remarkable success in continuous domains such as image generation, its direct application to discrete data has underperformed pure discrete formulations. To understand this gap, we introduce *token identifiability*, an analytical framework characterizing how Gaussian noise corrupts discrete data through two mechanisms: *discrete identity corruption* and *continuous rank degradation*. We reveal that these mechanisms scale differently with vocabulary size, creating a *temporal dissonance* that forces a tradeoff between learning continuous geometry and discrete structure. To address this, we propose **CANDI** (**C**ontinuous **AN**d **DI**screte diffusion), a hybrid framework that decouples discrete and continuous corruption, enabling simultaneous learning of both. This unlocks the benefits of continuous diffusion for discrete spaces: on controlled generation, CANDI enables classifier-based guidance with off-the-shelf classifiers through simple gradient addition; on text generation, CANDI outperforms masked diffusion at low NFE, demonstrating the value of learning continuous gradients for discrete spaces. We include the code on the project page: https://patrickpynadath1.github.io/candi-lander.

## 1. Introduction

Diffusion models have become a central tool in generative modeling, with score-based methods showing remarkable performance across a range of continuous domains (Sohl-Dickstein et al., 2015; Song & Ermon, 2019; Song et al., 2020a; Karras et al., 2022). Recent works have adapted diffusion to discrete settings by training with discrete noise instead of continuous Gaussian noise (Austin et al., 2023;

Hoogeboom et al., 2021; Lou et al., 2024; Sun et al., 2023). These pure discrete diffusion frameworks have achieved significant success, scaling to large language models and demonstrating strong performance across various text generation tasks (Shi et al., 2024; Ye et al., 2025; Nie et al., 2024). Moreover, such methods have generally outperformed direct applications of continuous diffusion to discrete spaces (Arriola et al., 2025; Sahoo et al., 2024).

This outperformance is not obvious *a priori*. Continuous diffusion, in principle, should have advantages: it learns a score function that jointly updates multiple positions and captures correlations among variables (Song et al., 2020b; Lou et al., 2024; Sahoo et al., 2024). Such parallel position updates are valuable, especially in compute-constrained settings, where few function evaluations are allowed. Yet, empirical results show continuous approaches have underperformed pure discrete approaches even at low NFE, contradicting these expectations.

In this paper, we investigate the cause of continuous diffusion's underperformance on discrete data and propose a simple yet effective solution to recover its benefits. We introduce *token identifiability* as a framework for analyzing how continuous Gaussian noise interacts with discrete structure, characterizing signal corruption along two axes: *discrete identity corruption*, which measures **if** an incorrect token is closest to the noisy latent, and *continuous rank degradation*, which measures **how many** incorrect tokens are closer to the noisy latent than the correct token. Both are crucial, as discrete identity corruption directly relates to learning conditional dependencies, while continuous rank degradation enables the learning of a continuous score function through denoising.

By studying the analytical forms for both discrete identity corruption and continuous rank degradation, we reveal a fundamental mismatch: discrete identity corruption rapidly destroys token identifiability as the number of categories grows, whereas continuous rank degradation remains stable. This creates a *temporal dissonance*: at the noise levels where the model can condition on identifiable tokens, the continuous denoising task is trivial as the signal has not degraded significantly. Conversely, at the noise levels where the continuous signal has degraded significantly, there are no identifiable positions for the model to condition on. As a

---

[1]Purdue University, West Lafayette, U.S.A [2]Google Deepmind. Correspondence to: Patrick Pynadath <ppynadat@purdue.edu>, Ruqi Zhang <ruqiz@purdue.edu>.

*Proceedings of the 43rd International Conference on Machine Learning*, Seoul, South Korea. PMLR 306, 2026. Copyright 2026 by the author(s).

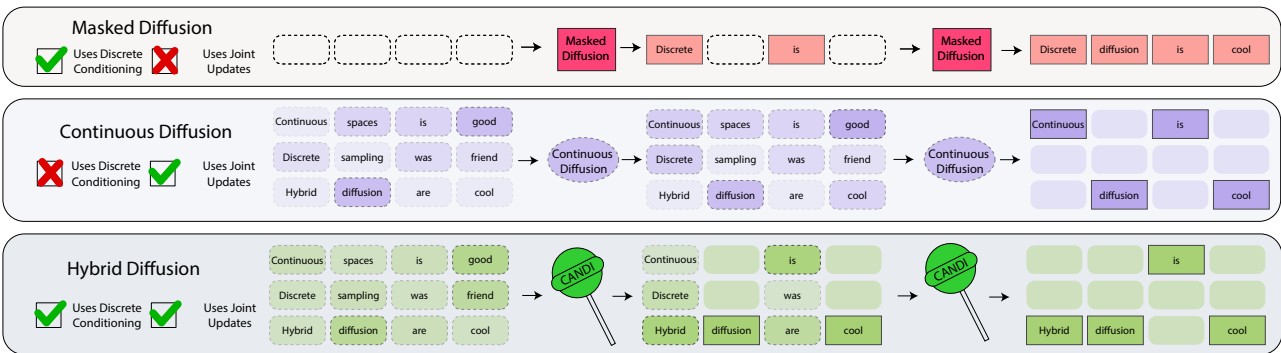

*Figure 1.* We provide a visual comparison of discrete, continuous, and hybrid diffusion. Discrete diffusion leverages discrete conditional structure, but lacks the ability to perform joint updates due to sampling each position independently. Continuous diffusion performs joint updates, but lacks discrete conditional structure. Our proposed hybrid diffusion, CANDI, can leverage both discrete conditional structure and joint updates.

result, the model cannot simultaneously learn the continuous score function and the discrete conditional structure.

To address this, we introduce **CANDI** (**C**ontinuous **And Di**screte diffusion), a framework that directly addresses the mismatch between discrete identity corruption and continuous rank degradation. CANDI uses discrete masking to preserve selected positions while applying Gaussian noise to others, decoupling discrete corruption from Gaussian noise dynamics. This separation paradoxically enables coordination: by controlling each mechanism independently, we ensure both forms of corruption scale gracefully together. As a result, CANDI takes the advantage of continuous geometry while still learning conditional dependencies. We demonstrate its effectiveness through classifier-based guidance, where continuous geometry enables guidance through simple gradient addition; and improved generative quality at low NFE, where continuous geometry allows for joint evolution across many positions simultaneously. In both settings, CANDI outperforms masked diffusion, verifying the benefits of continuous geometry for discrete spaces. We include a visual summary of our method and how it compares to other diffusion methods in Figure 1.

## 2. Related Works

**Discrete Diffusion Models** Discrete diffusion uses continuous-time Markov chains to define noising processes over categorical data (Campbell et al., 2022). Two main variants have emerged: masked diffusion, which corrupts tokens to a special mask token (Sahoo et al., 2024; Shi et al., 2024; Ou et al., 2024), and uniform diffusion, which samples from the uniform distribution over vocabulary (Lou et al., 2024; Schiff et al., 2024; Austin et al., 2023). While both approaches translate diffusion into a discrete form, they lack the benefits that Gaussian diffusion brings through continuous geometry.

**Continuous Diffusion for Discrete Data** Several works apply continuous diffusion to discrete sequences. Embedding-based methods inject Gaussian noise into learned token embeddings (Li et al., 2022; Dieleman et al., 2022; Strudel et al., 2022; Yuan et al., 2023) but must address degenerate embedding issues. Han et al. (2023) leverages Gaussian diffusion on the one-hot space, but relies on semi-autoregressive generation whereas our framework is fully non-autoregressive and focuses on the fundamental mismatch between discrete and continuous corruption mechanisms. Recently, Sahoo et al. (2025) shows a connection between uniform state diffusion and continuous Gaussian diffusion. However, they use this to motivate curriculum learning and distillation for uniform state diffusion, whereas we focus on understanding the fundamental barrier preventing Gaussian diffusion from being applicable to discrete spaces. Independent efforts have also recently explored hybrid discrete-continuous diffusion, which we discuss in Appendix A.

## 3. Preliminaries

Our goal is to learn a generative model for categorical sequences over a discrete vocabulary $\mathcal{V}$. We consider sequences of length $l$ where each position takes a value from $|\mathcal{V}|$ categories. This discrete $X$ can be represented as a sequence of $l$ one-hot vectors with dimension $|V|$, which we refer to as $\tilde{X}$. We distinguish between discrete tokens and the vector representations using $X$ and $\tilde{X}$ respectively.

**Masked Discrete Diffusion Process** We define the forward process as gradually interpolating between a ground truth distribution $P(X)$ and a degenerate distribution where all positions are replaced by a mask label indicating corruption (Shi et al., 2024; Sahoo et al., 2024; Lou et al., 2024).

$$q(X_t|X_0) = \text{Cat}\left(X_t; (1 - \alpha(t))X_0 + \alpha(t)M\right) \quad (1)$$

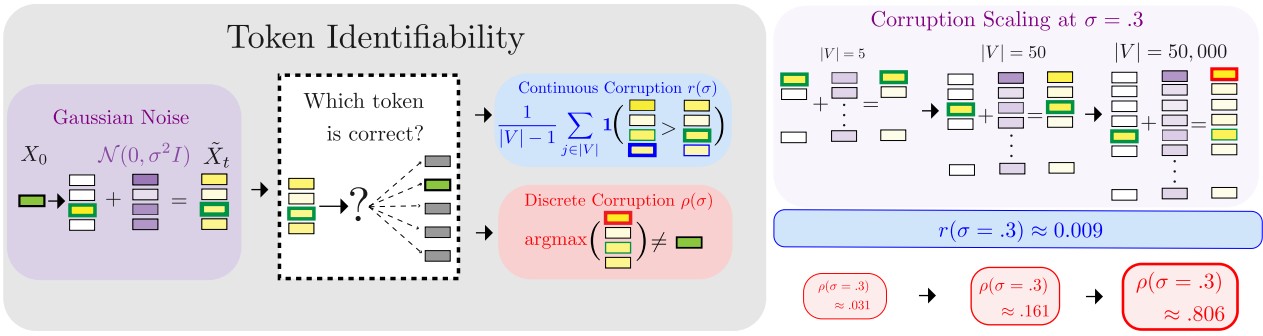

*Figure 2.* We visualize the two forms of corruption for token identifiability and visualize the asymmetric scaling with the number of categories $|V|$. Token identifiability, i.e., how the correct token relates to the noisy latent, is affected by Gaussian noise in two distinct mechanisms: (1) discrete corruption, determining whether the correct token can be obtained by taking the argmax of the noisy latent, and (2) continuous corruption, measuring how much signal from the correct token has degraded. While these two are different facets of the same underlying signal, they scale very differently with increases in vocabulary size $|V|$.

Here, $\alpha(t)$ is a monotonic and decreasing function. The reverse transitions are as follows:

$$q(X_s|X_t, X_0)$$
$$= \begin{cases} \text{Cat}(X_s; X_t), & \text{if } X_t \neq M \\ \text{Cat}\left(X_s; \frac{(1-\alpha_s)M + (\alpha_s - \alpha_t)X_0}{1-\alpha_t}\right) & \text{if } X_t = M. \end{cases} \quad (2)$$

In the case where $X_t = M$, the distribution $q(X_s|X_t)$ can be written as follows:

$$q(X_s|X_t) = \text{Cat}\left(X_s; \frac{(1-\alpha_s)M + (\alpha_s - \alpha_t)\mathbb{E}[X_0|X_t]}{1-\alpha_t}\right).$$

Thus learning $\mathbb{E}[X_0|X_t]$ is sufficient to simulate the reverse process.

**Continuous Diffusion Process** We define a diffusion process from $p(\tilde{X}_0)$ to a Gaussian prior as follows. The forward SDE is given by:

$$d\tilde{X}_t = g(t)dW_t.$$

This is a Variance-Exploding (VE) SDE, where $g(t)$ represents the diffusion coefficient (Song et al., 2020b). The corresponding reverse SDE is:

$$d\tilde{X}_t = -g(t)^2 \nabla_{\tilde{X}_t} \log p_t(\tilde{X}_t)dt + g(t)d\tilde{W}_t.$$

For deterministic sampling, we use the probability flow ODE, which removes the stochastic term:

$$d\tilde{X}_t = -\frac{1}{2}g(t)^2 \nabla_{\tilde{X}_t} \log p_t(\tilde{X}_t)dt. \quad (3)$$

The cumulative noise for time $t$ is defined as $\sigma(t)^2 = \int_0^t g(s)^2 ds$. We can define the score function in terms of the expectation $\mathbb{E}[\tilde{X}_0|\tilde{X}_t]$ and $\sigma(t)$, providing a convenient neural network parameterization:

$$\nabla_{\tilde{X}_t} \log p_t(\tilde{X}_t) = -\frac{\tilde{X}_t - \mathbb{E}[\tilde{X}_0|\tilde{X}_t]}{\sigma(t)^2}. \quad (4)$$

Here we observe that it is also the case that learning $\mathbb{E}[\tilde{X}_0|\tilde{X}_t]$ is sufficient to simulate the reverse process.

## 4. A New Lens: Token Identifiability under Gaussian Noise

Continuous diffusion models should theoretically excel at discrete sequence generation, as their continuous score functions enable simultaneous evolution of all positions in parallel. In practice, however, such models consistently underperform approaches based on discrete corruption kernels, despite the latter lacking continuous geometry. We resolve this paradox by introducing the concept of **token identifiability**: how the noisy latent relates to the correct token relative to all other incorrect tokens. Through this lens, we describe how Gaussian noise acts along two axes of corruption:

1. **Discrete Identity Corruption**: whether Gaussian noise discretely corrupts the categorical identity of the noisy latent

2. **Continuous Rank Degradation**: how Gaussian noise degrades the rank of the correct token relative to all other incorrect tokens

We analytically characterize both axes of signal corruption and reveal a **temporal dissonance** that occurs as the number of categories increases. This temporal dissonance hinders continuous diffusion from successfully modeling discrete categorical data. We include a visual summary in Figure 2.

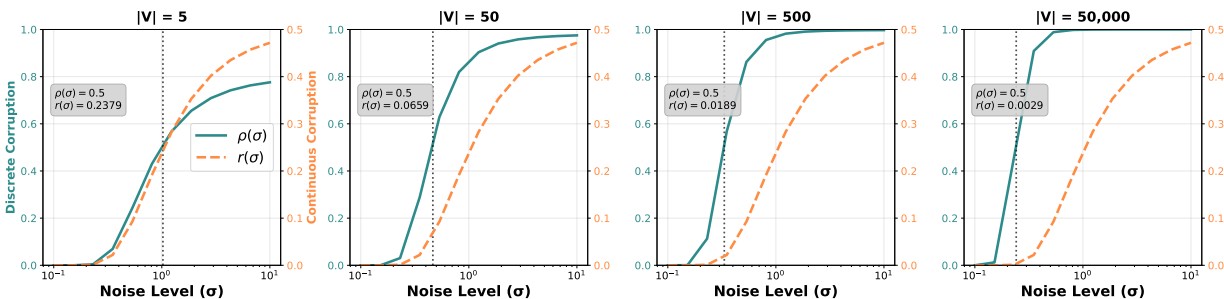

*Figure 3.* We plot $\rho(\sigma)$ and $r(\sigma)$ for different numbers of categories to illustrate the divergence between discrete identity corruption and continuous rank degradation. At larger values of $|V|$, even when half of the positions are argmax corrupted, the correct token index remain larger than 99% of the other coordinates. Thus when the model can learn conditional structure, it does not learn the continuous gradient; and when it learns the continuously denoise, it does not learn conditional structure.

### 4.1. Definitions

As our goal is to produce a sequence of discrete tokens, we rely on token identifiability as the primary notion of signal for the diffusion process. Token identifiability describes how the noisy latent $\tilde{X}_t$ relates to the correct token index $i$. This signal can be characterized from two complementary perspectives: discrete identity corruption (is the correct token identifiable?) and continuous rank degradation (how distinguishable is the correct token?). We now introduce the formal definitions and analytical expressions for both forms of signal corruption.

**Discrete Identity Corruption** Within the one-hot space, we define the discrete corruption of token identity as whether discretizing a noisy one-hot $\tilde{X}_t$ results in the correct token.

$$\rho(t) = P(\text{argmax}(\tilde{X}_0) \neq \text{argmax}(\tilde{X}_t)), \tilde{X}_t \sim \mathcal{N}(\tilde{X}_0, \sigma^2(t)I).$$

$$= \int_{-\infty}^{\infty} \left(1 - \left[\Phi\left(\frac{s}{\sigma(t)}\right)\right]^{|V|-1}\right) \cdot \mathcal{N}(s; 1, \sigma(t)^2)ds.$$

Here, $\Phi$ refers to the standard Gaussian CDF.

**Continuous Rank Degradation** While discrete corruption measures whether the noisy one-hot corresponds to an incorrect token, continuous rank degradation measures the fraction of incorrect tokens whose index values exceed that of the correct token—equivalently, the expected rank of the correct token among all tokens when sorted by the noisy one-hot's coordinate values. Given the correct token index $i$, we define continuous rank degradation as $r(t)$ as follows:

$$r(t) = \mathbb{E}_{\tilde{X}_t \sim \mathcal{N}(\tilde{X}_0, \sigma(t)^2 I)} \left[\frac{1}{|V|-1}\sum_{i \neq j} \mathbf{1}(\tilde{X}_t[j] > \tilde{X}_t[i])\right] \tag{5}$$

$$= \Phi\left(-\frac{1}{\sigma(t)\sqrt{2}}\right). \tag{6}$$

This can be interpreted as an analogue to the canonical SNR (signal-to-noise ratio) used in continuous domains. While SNR typically reflects *absolute* signal decay for the original continuous signal, our formulation reflects *relative* signal strength between the correct token $i$ and competing incorrect tokens $j$. We provide the derivation in Appendix B.2 along with an explanation on why this formulation is more natural for discrete domains.

### 4.2. Why Continuous Diffusion Fails on Discrete Data

Having defined both corruption metrics, we now discuss their role in training dynamics. Discrete identity corruption controls how the model learns conditional dependencies between tokens. Unlike images, where nearby pixels share values due to spatial smoothness (Dieleman, 2024), discrete sequences lack geometric continuity: conditionally related tokens such as "New" and "York" have entirely different identities. To capture such co-occurrence patterns, the model must see clean anchor tokens during training. Without them, it cannot learn discrete dependency structure.

Continuous rank degradation controls how the model learns the score function via continuous denoising, enabling coordinated refinement across positions. This gradient signal is the key strength of continuous diffusion: it allows multiple positions jointly evolve at once. Without access to a continuous score function, updates rely on independent conditional sampling, which leads to incoherent generations, especially at low NFE where many positions must be updated in parallel.

Effective discrete diffusion thus requires learning both conditional structure and continuous gradients. If only conditional structure is learned, the model cannot coordinate parallel updates; if only the continuous score is learned, it loses discrete dependency information. Only by combining both can the model jointly refine positions while maintaining coherence. However, as we demonstrate, Gaussian noising alone makes this impossible due to a temporal dissonance arising from discrete identity corruption's dependence on

vocabulary size.

**Temporal Dissonance** While both corruptions of token identity should be coordinated with each other, we observe that the discrete identity corruption scales problematically with the number of categories whereas the continuous rank degradation remains independent of the number of categories. As the number of categories increases, discrete identity corruption is exponentially dependent on $|V|$, as shown in (5). In contrast, continuous rank degradation is independent of $|V|$, as shown in (6). This creates a ***temporal dissonance***: the noise levels that preserve identifiable tokens no longer align with those required to learn a meaningful continuous score function. We visualize this in Figure 3, where we observe that this problematic scaling occurs for even moderate number of categories.

The implications of this dissonance are severe: under standard Gaussian noising for data with large number of categories, the noise levels that enable learning the continuous score function do not preserve enough identifiable tokens to learn conditional structure.

**Connection to Previous Continuous Diffusion** We hypothesize that this dilemma has necessitated the various augmentations that have been previously proposed to improve continuous diffusion over discrete spaces. In Appendix B.3, we contextualize prior techniques for Gaussian diffusion on discrete data in terms of our analysis, including self-conditioning, prefix completion, and random infilling. We also discuss how the temporal dissonance identified in this work relates to both embedding and latent diffusion in Appendix B.4 and B.6 respectively.

# 5. CANDI: Continuous And Discrete diffusion

To address the temporal dissonance between discrete identity corruption and continuous rank degradation, we introduce CANDI, a hybrid framework that integrates continuous and discrete diffusion. First, we introduce a structured noising kernel that decouples discrete corruption from continuous noise. This design eliminates the problematic $|V|$-dependent scaling of discrete corruption and sets both forms of corruption to be linear with respect to time, thus enabling a stable relation across all time steps. Then, we discuss the model parameterization and training procedure. Finally, we present an efficient inference algorithm that replaces the computationally expensive matrix multiplications required for large, noisy one-hot vectors with an approximate embedding lookup that closely follows the full continuous ODE dynamics.

## 5.1. Structured Noising Kernel

We seek a noising process that enables the model to condition on clean positions for learning the conditional structure

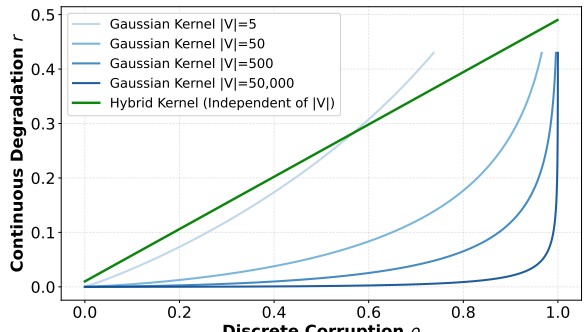

*Figure 4.* We demonstrate that our hybrid kernel eliminates the temporal dissonance induced by a large number of categories by decoupling discrete and continuous corruption. While the Gaussian kernel (blue lines) requires complete discrete corruption $\rho = 1$ to achieve meaningful continuous degradation at large vocabulary size, our hybrid approach maintains a linear relationship between $\rho$ and $r$ by explicitly controlling discrete corruption through a masking schedule.

of discrete data, while simultaneously injecting Gaussian noise to learn a continuous score function. To do this, we draw inspiration from masked diffusion to define a masking process that explicitly determines which positions to preserve as clean and which positions to corrupt with Gaussian noise. By decoupling discrete and continuous noising schedule, our structured noising kernel re-aligns continuous rank degradation and discrete identity corruption, avoiding the temporal dissonance that occurs with a large number of categories.

**Discrete Masking Process** We consider sequences of length $L$, where each position takes one of $|V|$ possible categories. We define a position-wise masking variable $M_t \in \{0,1\}^L$, where $M_t[i] = 1$ indicates that position $i$ remains clean and $M_t[i] = 0$ indicates that it is corrupted. The marginal distribution for each $M_t[i]$ is defined by a time-dependent keep rate $\alpha(t)$:

$$M_t[i] \sim \text{Bernoulli}(\alpha(t)), \ \alpha(t) = 1 - t. \quad (7)$$

This definition of $\alpha(t)$ follows the log-linear schedule introduced by Austin et al. (2021). As the corruption rate is simply $\rho(t) = 1 - \alpha(t) = t$, we remove the problematic $|V|$ dependence and enable the corruption rate to have a linear relation with time. At $t = 0$, all positions are clean ($M_0[i] = 1$), while at $t = 1$, all are corrupted ($M_1[i] = 0$). We further impose a carry-over masking constraint (Lou et al., 2024; Sahoo et al., 2024; Shi et al., 2024), ensuring that once a position is corrupted, it remains corrupted thereafter:

$$P(M_t[i] = 1 | M_s[i]) = \begin{cases} 0 & M_s[i] = 0 \\ \frac{\alpha(t)}{\alpha(s)} & M_s[i] = 1. \end{cases} \quad (8)$$

**Gaussian Noising Process** On the continuous side, we de-

fine a Gaussian diffusion process on $\mathbb{R}^{L \times |V|}$ with marginals

$$\tilde{X}_t \sim \mathcal{N}(\tilde{X}_0, \sigma(t)^2 I). \tag{9}$$

The noise level $\sigma(t)$ controls the degree of continuous signal degradation. We define a target degradation function $r^*(t) : [0, 1] \to [r_{\min}, r_{\max}]$ that specifies the desired fraction of incorrect token coordinates exceeding the correct token's value at each time step: $r^*(t) = (r_{\max} - r_{\min}) \cdot t + r_{\min}$. We constrain $r^*(t)$ to $[r_{\min}, r_{\max}]$ as continuous rank degradation $r(\sigma) \to 0.5$ only when $\sigma \to \infty$, which provides no learning signal. This definition of $r^*(t)$ ensures that continuous rank degradation maintains a linear relation with time. Given the target degradation rate $r^*(t)$, we compute the required noise level by inverting (6):

$$\sigma(t) = -\frac{1}{\Phi^{-1}(r^*(t))\sqrt{2}}. \tag{10}$$

As $r^*(t) \in (0, .5)$ for all $t \in [0, 1]$, $-\Phi^{-1}(u(t)) > 0$ and thus the derived inverse is always positive. This schedule can be interpreted as a specific instance of entropic schedules from Stancevic et al. (2025); Dieleman et al. (2022), where we define the information loss in terms of closeness to incorrect tokens v.s correct tokens.

**Hybrid Structured Noising Kernel** Given the distribution for the masking vector $M_t$ and the Gaussian diffusion, we now define the hybrid structured noising kernel to enable models to learn both discrete conditioning on clean "anchor" tokens and denoising of Gaussian corruption. We define the time marginals conditioned on clean data as follows:

$$X_t = X_0 \odot M_t + \tilde{X}_t \odot (1 - M_t). \tag{11}$$

We refer to this forward corruption kernel as $q_t(\cdot | X_0)$. Here, $M_t, \tilde{X}_t$ are sampled from the marginals defined in (8) and (9) respectively. Continuous rank degradation is controlled by $\tilde{X}_t$, which has $r(t)$ corrupted linearly with respect to time via the defined $r^*(t)$. Discrete identity corruption is controlled by $M_t$, which is also corrupted linearly with respect to time via $\alpha(t) = 1 - \rho(t)$. As both corruption forms vary linearly with time, this kernel enables the model to learn both the conditional structure and the continuous score function. It should be noted that this linear schedule was selected for simplicity, the optimal relation between $r^*$ and $\rho(t) = 1 - \alpha(t)$ is left for future work. Figure 4 illustrates how the behavior of $r(t)$ and $\rho(t)$ under this new structured kernel.

**Reverse Conditional Distribution** The reverse transition of the hybrid process is defined as:

$$P(X_s, M_s | X_t, M_t) =$$
$$\begin{cases} \delta(X_s - X_t) \cdot \delta(M_s - 1), & M_t = 1 \\ P_{\text{unmask}}, & M_t = 0, \text{ prob } p_u \\ P_{\text{denoise}}, & M_t = 0, \text{ prob } 1 - p_u \end{cases}$$
$$\tag{12}$$

where $p_u = \frac{\alpha(s) - \alpha(t)}{1 - \alpha(t)}$ and the terms are defined in the text.

A full derivation is provided in Appendix C.1.

$P_{\text{Gaussian}}(X_s | X_t)$ can be sampled by simulating the reverse ODE defined in (3). This hybrid approach applies continuous diffusion dynamics to discrete spaces while preserving discrete conditional structure. While other works (Sahoo et al., 2025) connect Gaussian and discrete diffusion through the argmax operation, they cannot leverage the full continuous dynamics during inference due to the loss of information after applying the argmax projection. Our hybrid approach is unique in that it combines both discrete corruption and continuous degradation into one noising process.

## 5.2. Model Parameterization and Training

**Model Parameterization** In order to simulate the reverse process of our structured corruption, the model must learn $P(X_0 | X_t)$ and $\nabla \log p_t(X_t)$. Given that $\nabla \log p_t(X_t)$ and the discrete transitions can be calculated from $\mathbb{E}[\tilde{X}_0 | \tilde{X}_t]$, learning $P_\theta(X_0 | X_t)$ allows for simulating both the discrete and continuous components of the reverse trajectory. Thus we adopt the objective for masked diffusion, which can be interpreted as a weighted cross entropy loss for learning $P_\theta(\cdot | X_t)$. For further details, see Appendix C.5.

## 5.3. Hybrid Inference

We introduce a hybrid inference algorithm which leverages both discrete conditional structure and continuous score function. We first describe how the ancestral update from masked diffusion is combined with the deterministic ODE update from Gaussian diffusion. We then introduce an approximate inference algorithm, which uses the sampling trick from Cetin et al. (2025) to avoid materializing the full noisy one-hot vector.

**Exact Inference** As a result of learning a continuous score function and the conditional discrete structure, our model is able to simultaneously select tokens to update while refining noisy positions. Given the current state $(\tilde{X}_t, M_t)$ and the next time step $s < t$, we first sample which positions will transition to a clean state:

$$M_s' \sim \text{Bernoulli}\left(\frac{\alpha(s) - \alpha(t)}{1 - \alpha(t)}\right).$$

For positions that transition to a clean state, we sample from the learned posterior $P_\theta(X_0 | X_t)$. For positions that remain corrupted, we apply the standard update formula for the reverse probability flow ODE. For more details, see Appendix C.2.

**Approximate Inference** The exact hybrid inference algorithm requires expensive matrix multiplications of size $|V| \times L \times B$ to form the noisy one-hot embeddings. In contrast, purely discrete diffusion methods perform efficient

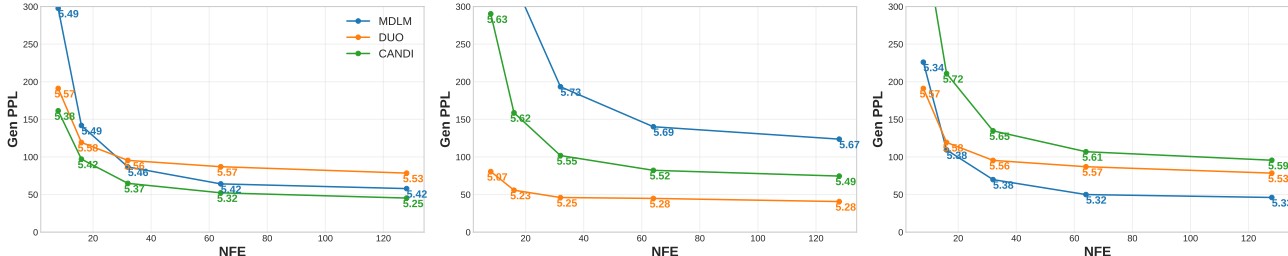

*Figure 5.* We demonstrate that single-point evaluations for perplexity and entropy can lead to misleading results. For each plot, we adjust the temperature of each method ($\tau \in [.875, 1.0]$) to achieve reasonable ranges of entropy (5.2-5.7, annotated along the curves) (Sahoo et al., 2025; Zheng et al., 2024). We show that small changes in this hyperparameter can produce entirely different relative performance rankings. We provide details on temperature settings for each plot in Appendix D.1.

embedding lookups without explicitly constructing one-hot representations.

We address this inefficiency by recognizing that our model effectively learns $P_\theta(\cdot|Y_t = W\tilde{X}_t)$. We follow Cetin et al. (2025) and employ a Monte Carlo approximation: we sample discrete tokens from $P_\theta(\cdot|Y_t)$ and use embedding lookup to approximate $\mathbb{E}[Y_0|W\tilde{X}_t]$ in a single step. This eliminates the need to materialize noisy one-hot vectors beyond initial prior sampling, thus achieving the advantages of continuous score functions while matching the efficiency of pure discrete methods. We include the algorithm along with additional details in Appendix C.3.

**Guidance** Previous work has explored classifier-based guidance for text diffusion models over text (Li et al., 2022; Schiff et al., 2024). As a result of using continuous geometry, our method naturally supports controllable generation through gradient-based guidance, without requiring diffusion-specific classifiers. Given a classifier $f_\phi : \mathcal{X} \to C$ over class labels $C$, we compute the gradient of the classifier with respect to the inputs for the desired class. We then add this gradient to the learned score function to compute the ODE update. For more details, see Appendix C.4.

# 6. Experimental Results

Our experiments verify two findings: (1) vocabulary size causes continuous diffusion's failure on discrete data through the temporal dissonance mechanism, and (2) CANDI resolves this while enabling improved low-NFE generation and gradient-based control. First, we introduce frontier analysis as our primary means of quantifying generative quality and provide a motivating example as to why frontier analysis is necessary. Next, we provide direct empirical support for the temporal dissonance analysis, demonstrating that continuous diffusion attains reasonable performance for small vocabularies but fails catastrophically at large vocabularies. We also show that CANDI avoids this temporal dissonance, enabling strong performance with a large number of categories. Finally, we demonstrate how

CANDI leverages continuous geometry to improve both low-NFE generation quality and enable controllable generation through simple gradient addition.

## 6.1. Why Frontier Analysis is Necessary

Throughout our experiments, we focus on measuring generative quality as opposed to likelihood evaluations. We avoid likelihood-based metrics as they assume infinitesimal steps ($dt \to 0$), whereas our focus is practical generation with a finite number of timesteps.

**Frontier Analysis** To compare generative quality across models, we extend the *entropy-perplexity frontier analysis*, which captures the trade-off between sample diversity (entropy) and fluency (perplexity) (Zheng et al., 2024). For each method, we vary the temperature used for sampling tokens, which results in different operating points of diversity v.s coherence. We compare methods by plotting the entire frontier for each method. We show why frontiers are necessary in Figure 11: despite all entropies being within a reasonable range, slight temperature changes can reverse relative rankings. For a more thorough discussion on this phenomenon, see Pynadath et al. (2026).

## 6.2. Empirical Validation of Temporal Dissonance

Our temporal dissonance hypothesis implies three behaviors of continuous diffusion on discrete data. ***First***, when the vocabulary is small enough, continuous diffusion should maintain reasonable performance. ***Second***, when the vocabulary is large, it should exhibit complete generative failure—either mode collapse or highly random generations. ***Third***, CANDI should remain competitive with discrete methods regardless of vocabulary size, since it disentangles discrete corruption from vocabulary size.

In this section, we use generative frontiers for both Text8 and OpenWebText to empirically verify all three predictions, validating the temporal dissonance hypothesis.

**Experimental Design** We use Text8 (Mahoney, 2011) and

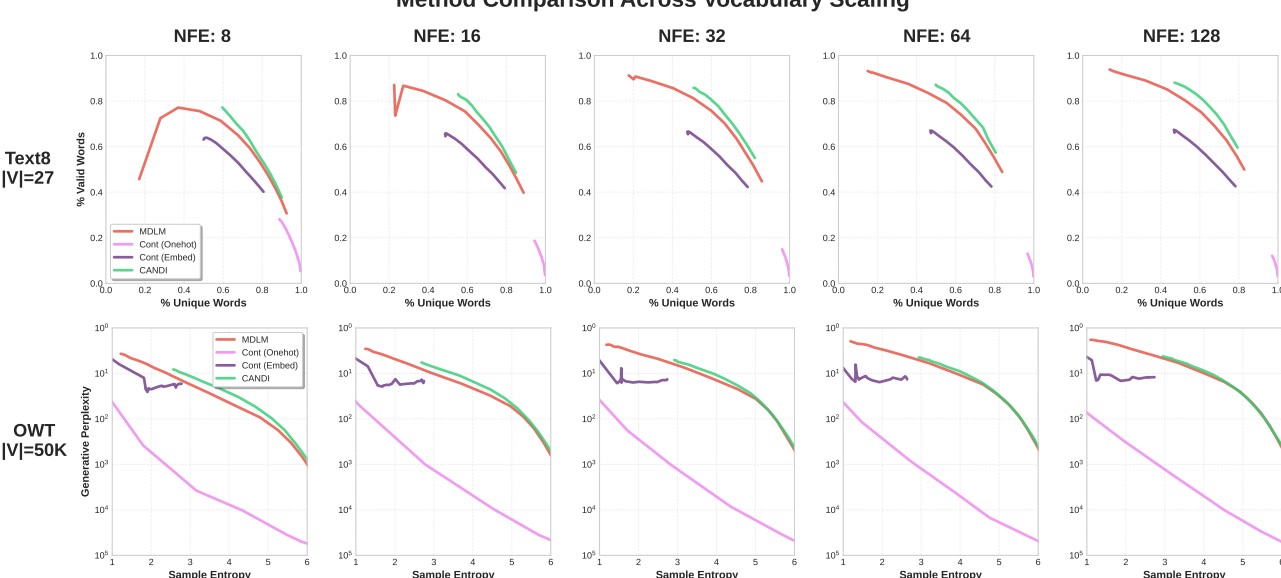

*Figure 6.* We include the generative frontier results on Text8 and OWT for One-hot diffusion (pink), embedding diffusion (purple), MDLM (red), and CANDI (green). We observe that pure continuous diffusion methods maintain reasonable frontiers on Text8 across all NFE but collapse on OWT. This aligns with the predicted behavior from our token identifiability framework. Furthermore, disentangling discrete corruption rate from vocabulary size achieves the desired stability, as CANDI matches or exceeds MDLM performance at all visualized NFE.

OpenWebText (Gokaslan et al., 2019) as examples of small and large vocabulary data respectively. We train two pure continuous diffusion models on each dataset: the first is a continuous diffusion model on the one-hot, following the SDE defined in (9); the second is an embedding diffusion model, using a VE SDE (Song et al., 2020b).

**Evaluation Metrics** To properly evaluate the generative ability of each model, we use diversity-coherence metrics specific to each dataset to ensure proper comparisons and visualize them in terms of frontiers. For Text8, we use the percentage of unique words and percentage of valid words to measure diversity and coherence respectively, as these are naturally suited towards character-based data. For OWT, we use the standard entropy and perplexity for diversity and coherence (Arriola et al., 2025; Sahoo et al., 2025). While metrics differ across datasets, we focus on how continuous diffusion's relative performance versus CANDI and MDLM changes with vocabulary size (Figure 6). For more details on experimental design and evaluation metrics, see Appendix D.2.

**Results** As shown in Figure 6, continuous diffusion methods maintain reasonable performance at small vocabularies but fail catastrophically at large vocabularies. On Text8, we observe that the generative frontiers of both one-hot diffusion and embedding diffusion are fairly close to that of MDLM and CANDI. On OWT, we observe that the frontier for one-hot diffusion is orders of magnitude worse than either MDLM or CANDI. Furthermore, embedding diffusion

appears mode collapsed due to the extremely low entropy and unstable behavior with temperature tuning.

Lastly, we note that CANDI does not suffer from these same defects, as it outperforms MDLM on both Text8 and OWT. This strongly supports the temporal dissonance hypothesis, as we observe both the failure that arises due to the mismatch and the improved performance once the mismatch is directly addressed.

### 6.3. Improved Low-NFE Generation Quality: OWT

**Experimental Design** By leveraging continuous geometry during inference, CANDI should achieve superior generation quality at low NFE compared to pure masked discrete diffusion, as the continuous gradient provides coordination across the entire sequence. We compare against MDLM (Sahoo et al., 2024) and DUO (Sahoo et al., 2025) to serve as representative masked diffusion and uniform diffusion baselines. For further training and experimental details, see Appendix D.3.

**Results** In Figure 7, we observe that CANDI has a strictly better frontier than masked diffusion up to 64 NFE, where it matches performance. CANDI also exhibits a strictly superior frontier than DUO as perplexity decreases past approximately 40. This demonstrates that leveraging both continuous score information and discrete conditional information enables superior generative quality at low NFE. We include further results in Appendix D.3.

**Method Comparison Across NFE**

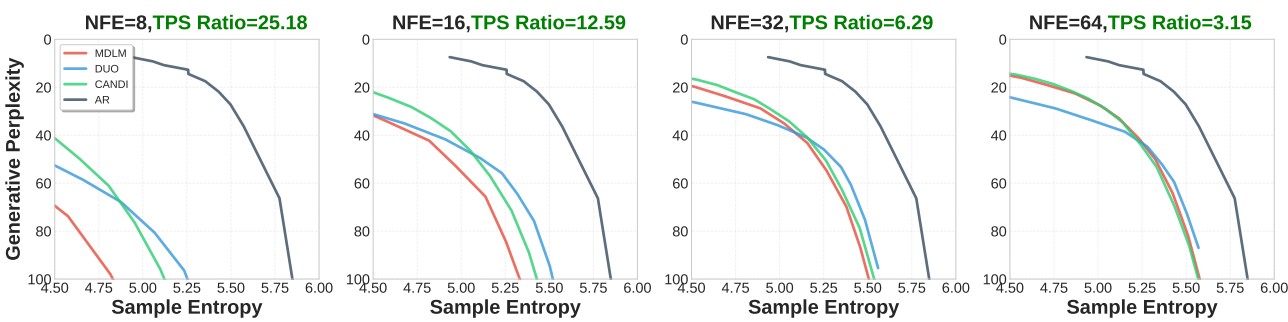

*Figure 7.* We show the entropy-perplexity frontiers for NFE=8, 16, 32, 64. Each subplot includes the TPS advantage (diffusion throughput/autoregressive throughput) to highlight the efficiency advantage of diffusion models at low NFE. CANDI outperforms MDLM up to 64 NFE and surpasses DUO as perplexity falls below 40.

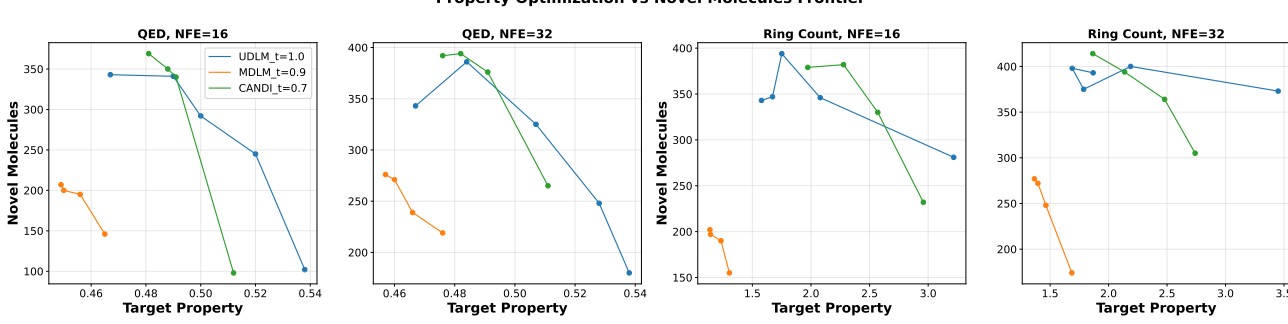

*Figure 8.* We compare the frontiers for MDLM, UDLM, and CANDI at NFE = 16 and 32 after applying classifier-based guidance and optimizing temperatures for each method. Different points along each frontier represent different guidance strengths for that specific temperature. Despite using an off-the-shelf classifier with no diffusion-specific training, CANDI is able to achieve a competitive frontier with UDLM.

### 6.4. Controlled Generation: QM9

**Experimental Design** Here, we evaluate CANDI's ability to perform controllable generation using *off-the-shelf* classifiers compared with prior discrete diffusion methods that use specialized diffusion classifiers. Following the QM9 guidance setup (Schiff et al., 2024), we compare CANDI against their proposed classifier-guided versions of UDLM and MDLM. We train classifiers to predict QED and ring count, and evaluate performance across a range of classifier guidance strengths. For more details, see Appendix D.4.

**Evaluation Metrics** For each method, we report the best Pareto frontiers in terms of the target attribute (coherence) and the number of unique, valid molecules (diversity) obtained after a grid search over temperature values. Additional training details and temperature sweep results are provided in Appendix D.4, and the QED and ring-count frontiers are shown in Figure 8.

**Results** We observe that CANDI strictly dominates MDLM and achieves competitive performance against UDLM, without requiring a specialized classifier. This demonstrates that hybrid diffusion naturally enables classifier-based guidance

without the need for specialized classifiers.

## 7. Conclusion

In this work, we study the challenges and opportunities of applying continuous diffusion models to discrete data. First, we introduce token identifiability, a framework for characterizing the effect of continuous noise on discrete data. Using this framework, we identify the root cause of continuous diffusion's underperformance on discrete data as a temporal dissonance between discrete and continuous corruption of token identity. We propose CANDI, a hybrid approach that decouples these corruption mechanisms, enabling simultaneous learning of discrete conditional structure and continuous score functions. We empirically verify the existence of this temporal dissonance and show that CANDI resolves it, allowing continuous diffusion to succeed in discrete spaces. We hope this work motivates further development of unified diffusion perspectives that can be applied across both discrete and continuous domains.

## Acknowledgements

This research is supported in part by NSF IIS-2508145 and Amazon Research Award.

## Impact Statement

This paper presents work whose goal is to advance the field of Machine Learning. There are many potential societal consequences of our work, none which we feel must be specifically highlighted here.

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

# Appendix

## A. Concurrent Works

Independently, other works have demonstrated the benefits of hybrid discrete-continuous diffusion. CADD (Zheng et al., 2025) augments masked diffusion using continuous embedding diffusion, motivated by the "information void" of pure masked diffusion. Additionally, CCDD (Zhou et al., 2025) argue that continuous diffusion is more expressive than discrete diffusion, and use this to justify combining discrete and continuous diffusion into a joint multi-modal framework with a factorized reverse process. We delineate the core differences between our contributions below:

- We explain why continuous diffusion has underperformed and how hybrid methods avoid this problem. While Zhou et al. (2025) discusses the expressivity-trainability tradeoff, we introduce a theoretical framework — token identifiability and temporal dissonance—that explains when continuous diffusion fails on categorical data and why hybrid methods avoid this issue.

- We focus on how continuous diffusion enables superior low-NFE generation quality and simple classifier-based guidance. These are unique benefits of hybrid diffusion not discussed in Zheng et al. (2025); Zhou et al. (2025).

- We extend frontier analysis as our primary means of quantifying generative quality, providing a general evaluation method applicable beyond hybrid diffusion.

## B. Theoretical Analysis for Discrete Spaces

### B.1. Alternative Derivation of Discrete Identity Corruption

We note that Sahoo et al. (2025) already provides the result for (5). We offer an alternative derivation that may provide additional intuition for discrete identity corruption $\rho(t)$.

**Discrete Corruption Rate at $t = 0$**   At $t = 0$, assume we have that $\tilde{X}_0$ is a one-hot for $V$ categories.

We can define the identity corruption $\rho(t)$ as follows:

$$\rho(t) = P(\text{argmax}(\tilde{X}_0) \neq \text{argmax}(\tilde{X}_t)), \ \tilde{X}_t \sim \mathcal{N}(\tilde{X}_0, \sigma^2(t)I).$$

Assume that the correct token index is $i$. We can use law of total probabilities to express $\rho(t)$ as follows:

$$\rho(t) = P(\exists j \neq i, \ \tilde{X}_t[j] > \tilde{X}_t[i])$$
$$= 1 - P(\forall j \neq i, \ \tilde{X}_t[j] < \tilde{X}_t[i]).$$

We approach this problem by marginalizing over values $s$ that the correct token index for $\tilde{X}_t[i]$ can realize.

$$\rho(t) = \int_{-\infty}^{\infty} (1 - P(\forall j \neq i, \ \tilde{X}_t[j] < \tilde{X}_t[i] \mid \tilde{X}_t[i] = s)) \, p(\tilde{X}_t[i] = s) ds.$$

As $\tilde{X}_t[j] \sim \mathcal{N}(0, \sigma(t)^2 I)$, we obtain the following via standard Gaussian CDF properties:

$$P(\tilde{X}_t[j] < \tilde{X}_t[i] | \tilde{X}_t[i] = s) = \Phi\left(\frac{s}{\sigma(t)}\right). \tag{13}$$

All the incorrect token indices are sampled from Gaussians with mean 0 and equal variance. Thus, the probability that $V - 1$ indices fail to overtake the correct token index is as follows:

$$P(\forall j \neq i, \ \tilde{X}_t[j] < \tilde{X}_t[i] \mid \tilde{X}_t[i] = s) = \left(\Phi\left(\frac{s}{\sigma(t)}\right)\right)^{|V|-1}. \tag{14}$$

Substituting this into the original integral and using the Gaussian density function for $\tilde{X}_t[i] = s$ where $\tilde{X}_t[i] \sim \mathcal{N}(1, \sigma(t)^2)$, we obtain the integral from (5):

$$\rho(t) = \int_{-\infty}^{\infty} \left(1 - \left[\Phi\left(\frac{s}{\sigma(t)}\right)\right]^{|V|-1}\right) \cdot \mathcal{N}(s; 1, \sigma(t)^2) ds \tag{15}$$

**Discrete Identity Corruption for** $t > 0$  If we consider the forward time marginal for some $\tau > 0$ with a general $\tilde{X}_\tau$, the corruption rate for the argmax-projected Gaussian diffusion becomes state dependent. For clarity, we will still refer to the correct token index as $i$ and use $j$ to represent incorrect token indices.

First, we observe that the probability of any incorrect token index $j$ overtaking the correct index $i$ when it realizes value $s$ can be written as follows:

$$P(\tilde{X}_t[j] < \tilde{X}_t[i]|\tilde{X}_t[i] = s) = \Phi\left(\frac{s - \tilde{X}_\tau[j]}{\sigma(t)}\right).$$

Without assumptions on $i, j$, this cannot be simplified further as $\tilde{X}_\tau[j]$ can take on any value with non-zero probability.

Second, as the $V - 1$ incorrect indices may have different means, the probability that none overtake the correct token index is defined as follows:

$$P(\forall j \neq i, \ \tilde{X}_t[j] < \tilde{X}_t[i] \mid \tilde{X}_t[i] = s) = \prod_{j \neq i} \Phi\left(\frac{s - \tilde{X}_\tau[j]}{\sigma(t)}\right). \tag{16}$$

Finally, since we no longer know the original value of $\tilde{X}_\tau[i]$, the PDF of the distribution for the correct token index cannot be simplified past $\mathcal{N}(\tilde{X}_t[i], \sigma(t)^2)$. Thus, the final corruption rate is as follows:

$$\rho(t, \tilde{X}_\tau) = \int_{-\infty}^{\infty} \left(1 - \prod_{j \neq i} \Phi\left(\frac{s - \tilde{X}_\tau[j]}{\sigma(t)}\right)\right) \cdot \mathcal{N}(s; \tilde{X}_\tau[i], \sigma(t)^2) ds$$

As the argmax of the Gaussian diffusion for $\tau > 0$ depends on continuous magnitudes that simply do not exist in the discrete domain, the corruption dynamics for the argmax-projected Gaussian diffusion and the discrete uniform state diffusion diverge for any time $t$ past 0. This means that DUO (Sahoo et al., 2025) does not fully leverage the benefits of Gaussian diffusion.

### B.2. Continuous Rank Degradation

Here we provide a derivation for the continuous rank degradation $r(t)$. We assume that the correct token index is $i$, and we denote the incorrect token indices as $j$. We first write the definition of $r(t)$:

$$r(t) = \mathbb{E}_{\tilde{X}_t \sim \mathcal{N}(\tilde{X}_0, \sigma(t)^2 I)}\left[\frac{1}{|V| - 1} \sum_{i \neq j} \mathbf{1}(\tilde{X}_t[j] < \tilde{X}_t[i])\right].$$

Next, we apply the linearity of expectations:

$$\begin{aligned}
r(t) &= \frac{1}{|V| - 1} \sum_{i \neq j} \mathbb{E}_{\tilde{X}_t \sim \mathcal{N}(\tilde{X}_0, \sigma(t)^2 I)}\left[\mathbf{1}(\tilde{X}_t[j] > \tilde{X}_t[i])\right] \\
&= \frac{1}{|V| - 1} \sum_{i \neq j} P(\tilde{X}_t[j] > \tilde{X}_t[i]).
\end{aligned}$$

By definition, $\tilde{X}_t[i] \sim \mathcal{N}(1, \sigma(t)^2)$ and $\tilde{X}_t[j] \sim \mathcal{N}(0, \sigma(t)^2)$. Thus for $j \neq i$, we have the following:

$$\tilde{X}_t[j] - \tilde{X}_t[i] \sim \mathcal{N}(0 - 1, \sigma(t)^2 + \sigma(t)^2) = \mathcal{N}(-1, 2\sigma(t)^2).$$

As a result, we have the following:

$$P(\tilde{X}_t[j] > \tilde{X}_t[i]) = P(\tilde{X}_t[j] - \tilde{X}_t[i] > 0) = \Phi\left(-\frac{1}{\sigma(t)\sqrt{2}}\right).$$

We are also given that all $|V| - 1$ are i.i.d. Thus we have the following:

$$r(t) = \frac{1}{|V| - 1} \cdot (|V| - 1) \cdot \Phi\left(-\frac{1}{\sigma(t)\sqrt{2}}\right) = \Phi\left(-\frac{1}{\sigma(t)\sqrt{2}}\right),$$

which is the result shown in (6).

**Relation to Continuous SNR**   This continuous degradation metric can be viewed as an analogue to the canonical SNR (signal-to-noise ratio) typically used in continuous domains. SNR is typically defined as the ratio of the original continuous signal over the variance of the added noise (Kingma et al., 2021):

$$SNR(t) = \frac{a(t)}{\sigma(t)^2}.$$

Here, $a(t)$ is the scaling of the original signal in a VP SDE. For a VE SDE, it would simply be $1$. While this formulation captures the continuous decay of signal magnitude, it does not reflect how categorical information deteriorates. In discrete domains with a finite vocabulary, what matters is *not the absolute signal strength* but the *relative signal strength* of the correct token index to incorrect token indices. Our formulation captures this effect by directly measuring how many incorrect tokens become more prominent than the original correct token in the noisy latent.

This limitation of standard SNR also appears in embedding diffusion. Even if embeddings have large norms and are corrupted by only small amounts of Gaussian noise (yielding a high SNR), the underlying categorical information may still be unrecoverable if the embeddings have collapsed—i.e., different tokens occupy nearly identical regions of the embedding space (Nguyen et al., 2024; Gao et al., 2022; Dieleman et al., 2022). In such cases, the SNR suggests that the signal is easily recoverable, yet the model's uncertainty over token identity remains high. Our identifiability-based formulation addresses this discrepancy by quantifying recoverability in terms of relative rank rather than absolute magnitude, providing a more faithful measure of information loss for discrete domains.

**Empirical Validation**   Here we demonstrate that the derived expressions for discrete identity corruption $\rho(t)$ and continuous rank degradation $r(t)$ align with empirical simulation results. We test 5,000 points at variance levels $\sigma^2$ logarithmically spaced between .1 and 10 for vocabulary sizes $5, 50, 500, 50,000$. We include the results in Figure 9. We observe that the empirical results closely track the analytical expressions, confirming that discrete identity corruption is exponentially dependent on vocabulary size while continuous rank degradation is independent of vocabulary size.

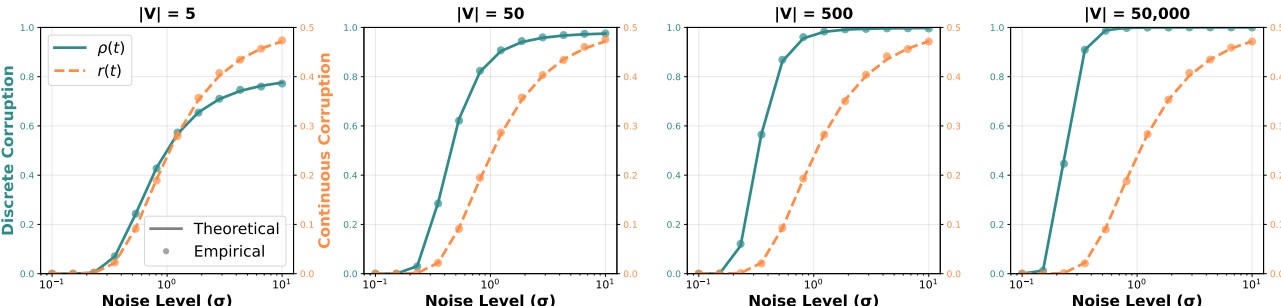

*Figure 9.* The empirical results against the analytical expressions for both discrete and continuous corruption. We observe that the empirical results closely align with the analytical predictions.

## B.3. Connection to Previous Continuous Diffusion Approaches

We discuss how our analysis relates to prior work that applies continuous diffusion to discrete spaces.

### B.3.1. CONTINUOUS DIFFUSION FOR SMALL VOCABULARY

Our analysis suggests that continuous diffusion would be performant for smaller vocabulary sizes. This is supported by the findings of Chen et al. (2022), where they demonstrate Gaussian diffusion performs well on binary data, which can be viewed as categorical with $|V| = 2$. While they do introduce self-conditioning, their extensive ablations show that

self-conditioning acts as an improvement on top of already viable performance. This aligns with our analysis of how discrete identity corruption and continuous rank degradation behave roughly similarly for a small number of categories, thus enabling the model to directly learn continuous and discrete structure.

### B.3.2. CONTINUOUS DIFFUSION FOR CATEGORICAL SPACES

We argue that continuous diffusion suffers from a larger number of categories due to the divergence between discrete identity corruption and continuous rank degradation. Here we briefly summarize solutions uncovered from prior works and how they align with our findings.

**Self-conditioning**    Many continuous diffusion methods have applied the self-conditioning mechanism from Chen et al. (2022) to improve performance (Dieleman et al., 2022; Strudel et al., 2022; Li et al., 2022; Gulrajani & Hashimoto, 2023). Self-conditioning enables the model to use its prior predictions to inform current predictions.

Through our framework, we can interpret self-conditioning as providing an alternative mechanism for learning conditional dependencies. While our method explicitly preserves clean token positions as conditioning signals, self-conditioning can be interpreted as using the model's own previous predictions as pseudo-clean anchors. In both cases, the model conditions on relatively confident token predictions (either truly clean or previously predicted) to denoise other positions.

This connection suggests that self-conditioning may have been empirically successful because it addresses the lack of identifiable conditioning signals in pure continuous diffusion. This complementary interpretation does not diminish the original insights but rather enriches our understanding of why certain architectural choices proved particularly effective for discrete domains.

**Semi-Autogressive Generation, Prefix Completion, and Infilling**    Prior works have also applied masked prefix training / random infilling (Dieleman et al., 2022; Strudel et al., 2022). While this approach provides the model with clean inputs to condition on, it does not directly fix the problematic behavior of the Gaussian kernel with respect to discrete identity corruption. Interestingly, Dieleman et al. (2022) discovers that training the diffusion model with random masks and prefix masks enables superior performance on prefix completion as opposed to training with only prefix masks.

While this may seem counterintuitive at first, our framework provides a natural explanation: random masking enables the model to learn conditional dependencies across the entire sequence, not just autoregressive left-to-right dependencies. While prefix masking teaches only sequential dependencies, random masking provides a learning signal for bidirectional conditional structure. As continuous diffusion does not follow auto-regressive generation and refine the entire sequence at each step, learning a bidirectional conditional structure is crucial for coherence.

Our token identifiability framework offers a unifying lens for understanding why certain architectural choices (masking, self-conditioning) proved particularly effective for discrete domains, potentially informing future design decisions.

### B.4. Extension to Embedding Diffusion

**Token Identity**    While we choose diffusion on the one-hot space as continuous and discrete corruption of token identity are easier to quantify, similar definitions also hold for embedding diffusion. First, we derive the embedding win rate, or the probability of original token $i$ being correctly identified from the noisy embedding $Y_t$ over some arbitrary incorrect token $j$. We derive the analytical expressions for both dot product similarity and $l_2$ distance. We then show how this leads to analogous definitions of discrete and continuous corruption of token identifiability, which reveal the same asymmetrical scaling with vocabulary size.

**Embedding Win Rate for Dot Product Similarity**    Assume we have some initial embedding vector $Y_0^{(i)}$ corresponding to token $i$ from a embedding table of vectors $Y_0^{(1)}, Y_0^{(2)}, \ldots Y_0^{(|V|)}$. We define $Y_t^{(i)} = Y_0^{(i)} + \sigma(t) \cdot \epsilon$, where $\epsilon \sim \mathcal{N}(0, I)$. For some arbitrary token $j$ with embedding $Y_0^{(j)}$, we define the "win-rate" $\omega(i, j, t)$ as follows:

$$\omega(i, j, t) = P(Y_t^{(i)} \cdot Y_0^{(i)} > Y_t^{(i)} \cdot Y_0^{(j)}). \tag{17}$$

This is simply the probability that the dot product between the noisy embedding and the original embedding is greater than the dot product between the noisy embedding and some incorrect embedding. Note that this cannot be computed in closed

form without knowledge of the specific embedding table. However, this probability can be written in terms of a Gaussian CDF. We derive this as follows:

$$\omega(i,j,t) = P(Y_t^{(i)} \cdot Y_0^{(i)} > Y_t^{(i)} \cdot Y_0^{(j)})$$
$$= P\left((Y_0^{(i)} \cdot Y_0^{(i)} + \sigma(t)\left(\epsilon \cdot Y_0^{(i)}\right) > Y_0^{(i)} \cdot Y_0^{(j)} + \sigma(t)\left(\epsilon \cdot Y_0^{(j)}\right)\right)$$
$$= P\left(\frac{1}{\sigma(t)}\left(||Y_0^{(i)}||_2^2 - Y_0^{(i)} \cdot Y_0^{(j)}\right) > \epsilon \cdot \left(Y_0^{(j)} - Y_0^{(i)}\right)\right).$$

Since $\epsilon \sim \mathcal{N}(0, I)$, the dot product on the right follows a single variable Gaussian distribution defined as follows:

$$\epsilon \cdot \left(Y_0^{(j)} - Y_0^{(i)}\right) \sim \mathcal{N}\left(0, ||Y_0^{(j)} - Y_0^{(i)}||_2^2\right)$$
$$\implies \frac{\epsilon \cdot \left(Y_0^{(j)} - Y_0^{(i)}\right)}{||Y_0^{(j)} - Y_0^{(i)}||_2^2} \sim \mathcal{N}(0, I).$$

As a result, we have the following:

$$P\left(\frac{1}{\sigma(t)}\left(||Y_0^{(i)}||_2^2 - Y_0^{(i)} \cdot Y_0^{(j)}\right) > \epsilon \cdot \left(Y_0^{(j)} - Y_0^{(i)}\right)\right) = P\left(\frac{\left(||Y_0^{(i)}||_2^2 - Y_0^{(i)} \cdot Y_0^{(j)}\right)}{\sigma(t)||Y_0^{(j)} - Y_0^{(i)}||_2^2} > \frac{\epsilon \cdot \left(Y_0^{(j)} - Y_0^{(i)}\right)}{||Y_0^{(j)} - Y_0^{(i)}||_2^2}\right)$$
$$= \Phi\left(-\frac{||Y_0^{(i)}||_2^2 - Y_0^{(i)} \cdot Y_0^{(j)}}{\sigma(t)||Y_0^{(j)} - Y_0^{(i)}||_2^2}\right).$$

Thus the win rate $\omega(i,j,t)$ can be computed as follows:

$$\omega(i,j,t) = \Phi\left(-\frac{||Y_0^{(i)}||_2^2 - Y_0^{(i)} \cdot Y_0^{(j)}}{\sigma(t)||Y_0^{(j)} - Y_0^{(i)}||_2^2}\right). \tag{18}$$

**Embedding Win Rate for L2 Distance**    It is also possible to derive a win-rate for when $l_2$ distance is used to determine how continuous embeddings map to discrete tokens.

$$\omega(i,j,t) = P(\|Y_t^{(i)} - Y_0^{(i)}\|_2^2 < \|Y_t^{(i)} - Y_0^{(j)}\|_2^2)$$
$$= P(\|Y_t^{(i)}\|_2^2 - 2 \cdot Y_t^{(i)} \cdot Y_0^{(i)} + \|Y_0^{(i)}\|_2^2 < \|Y_t^{(i)}\|_2^2 - 2 \cdot Y_t^{(i)} \cdot Y_0^{(j)} + \|Y_0^{(j)}\|_2^2)$$
$$= P(2 \cdot Y_t^{(i)} \cdot (Y_0^{(j)} - Y_0^{(i)}) < \|Y_0^{(j)}\|_2^2 - \|Y_0^{(i)}\|_2^2)$$
$$= P(2(Y_0^{(i)} + \sigma(t)\epsilon) \cdot (Y_0^{(j)} - Y_0^{(i)}) < \|Y_0^{(j)}\|_2^2 - \|Y_0^{(i)}\|_2^2)$$
$$= P(2Y_0^{(i)} \cdot (Y_0^{(j)} - Y_0^{(i)}) + 2\sigma(t)\epsilon \cdot (Y_0^{(j)} - Y_0^{(i)}) < \|Y_0^{(j)}\|_2^2 - \|Y_0^{(i)}\|_2^2)$$
$$= P\left(2\sigma(t)\epsilon \cdot (Y_0^{(j)} - Y_0^{(i)}) < \|Y_0^{(j)}\|_2^2 - \|Y_0^{(i)}\|_2^2 - 2Y_0^{(i)} \cdot (Y_0^{(j)} - Y_0^{(i)})\right)$$
$$= P\left(2\sigma(t)\epsilon \cdot (Y_0^{(j)} - Y_0^{(i)}) < \|Y_0^{(j)}\|_2^2 - \|Y_0^{(i)}\|_2^2 - 2Y_0^{(i)} \cdot Y_0^{(j)} + 2\|Y_0^{(i)}\|_2^2\right)$$
$$= P\left(2\sigma(t)\epsilon \cdot (Y_0^{(j)} - Y_0^{(i)}) < \|Y_0^{(j)}\|_2^2 + \|Y_0^{(i)}\|_2^2 - 2Y_0^{(i)} \cdot Y_0^{(j)}\right)$$
$$= P\left(2\sigma(t)\epsilon \cdot (Y_0^{(j)} - Y_0^{(i)}) < \|Y_0^{(j)} - Y_0^{(i)}\|_2^2\right).$$

Since $\epsilon \sim \mathcal{N}(0, I)$, we have:

$$\epsilon \cdot (Y_0^{(j)} - Y_0^{(i)}) \sim \mathcal{N}\left(0, \|Y_0^{(j)} - Y_0^{(i)}\|_2^2\right)$$
$$\implies \frac{\epsilon \cdot (Y_0^{(j)} - Y_0^{(i)})}{\|Y_0^{(j)} - Y_0^{(i)}\|_2} \sim \mathcal{N}(0, 1).$$

Therefore:

$$\omega(i,j,t) = P\left(\frac{2\sigma(t)\epsilon \cdot (Y_0^{(j)} - Y_0^{(i)})}{\|Y_0^{(j)} - Y_0^{(i)}\|_2} < \frac{\|Y_0^{(j)} - Y_0^{(i)}\|_2^2}{\|Y_0^{(j)} - Y_0^{(i)}\|_2}\right)$$

$$= P\left(\frac{2\sigma(t)\epsilon \cdot (Y_0^{(j)} - Y_0^{(i)})}{\|Y_0^{(j)} - Y_0^{(i)}\|_2} < \|Y_0^{(j)} - Y_0^{(i)}\|_2\right)$$

$$= \Phi\left(\frac{\|Y_0^{(j)} - Y_0^{(i)}\|_2}{2\sigma(t)}\right).$$

**Discrete Identity Corruption for Embedding Diffusion**   We show how to extend the idea of argmax corruption to continuous embedding diffusion.

For a given embedding table $Y_0^{(1)}, Y_0^{(2)} \ldots Y_0^{(|V|)}$, let us use $\Lambda$ to represent the set of all possible events. Similar to (5), we define the discrete embedding corruption rate as follows:

$$\rho_\omega(i,t) = P\left(\max_{j \in V, j \neq i} d\left(Y_t^{(i)}, Y_0^{(j)}\right) > d\left(Y_t^{(i)}, Y_0^{(i)}\right)\right). \tag{19}$$

We use $d$ to denote some function that reflects alignment between the two vectors, with higher values corresponding to higher alignment. We choose to keep in general the following derivation, as the specific choice of alignment or distance metric does not depend on the specific choice of alignment.

$$P\left(\max_{j \in V, j \neq i} d\left(Y_t^{(i)}, Y_0^{(j)}\right) > d\left(Y_t^{(i)}, Y_0^{(i)}\right)\right) = 1 - P\left(\max_{j \in V, j \neq i} d\left(Y_t^{(i)}, Y_0^{(j)}\right) \leq d\left(Y_t^{(i)}, Y_0^{(i)}\right)\right). \tag{20}$$

Define the event $A_t(i,j)$ to be the event where the embedding vector for $j$ is less aligned, or further from the noisy latent than the original embedding vector for token $i$. More formally,

$$A_t(i,j) := \left\{d\left(Y_t^{(i)}, Y_0^{(j)}\right) \leq d\left(Y_t^{(i)}, Y_0^{(i)}\right)\right\}. \tag{21}$$

This lets us express the corruption rate as follows:

$$P\left(\max_{j \in V, j \neq i} d\left(Y_t^{(i)}, Y_0^{(j)}\right) > d\left(Y_t^{(i)}, Y_0^{(i)}\right)\right) = 1 - P\left(\bigcap_{j \neq i} A_t(i,j)\right). \tag{22}$$

We assume that for a fixed $i$ and varying $j$, all corruption events $A_t(i,j)$ are independent of each other. In other words, the probability that $A_t(i,j)$ occurs does not yield any information on whether $A_t(i,k)$ occurs for $i \neq k$. This allows us to simplify the expression as follows:

$$1 - P\left(\bigcap_{j \neq i} A_t(i,j)\right) = 1 - \prod_{j \neq i} P\left(A_t(i,j)\right) \tag{23}$$

$$= 1 - \prod_{j \neq i} \omega(i,j,t) \tag{24}$$

$$= 1 - \left(\omega_{geo}(i,t)\right)^{|V|-1}. \tag{25}$$

We use $\omega_{geo}(i,t)$ to refer to the geometric mean of $\omega(i,j,t)$ over $j \in V, j \neq i$. This yields the final formula:

$$P\left(\max_{j \in V, j \neq i} d\left(Y_t^{(i)}, Y_0^{(j)}\right) > d\left(Y_t^{(i)}, Y_0^{(i)}\right)\right) = 1 - \left(\omega_{geo}(i,t)\right)^{|V|-1}. \tag{26}$$

We observe that discrete embedding corruption is exponentially dependent on the size of the vocabulary, similar to the argmax corruption rate derived in (5).

**Continuous Rank Degradation for Embedding Diffusion**   We define the continuous rank degradation for embedding diffusion as the expected fraction of incorrect embeddings that are more similar to the noisy embedding than the correct embedding. More formally,

$$r_\omega(i,t) = \mathbb{E}_{Y_t^{(i)}} \left[ \sum_{j \in V, j \neq i} \mathbf{1}\left( d\left(Y_t^{(i)}, Y_0^{(i)}\right) < d\left(Y_t^{(i)}, Y_0^{(j)}\right) \right) \right]. \tag{27}$$

This can be computed as follows:

$$\begin{aligned} r_\omega(i,t) &= \frac{1}{|V|-1} \mathbb{E}_{Y_t^{(i)}} \left[ \sum_{j \in V, j \neq i} \mathbf{1}\left( d\left(Y_t^{(i)}, Y_0^{(i)}\right) < d\left(Y_t^{(i)}, Y_0^{(j)}\right) \right) \right] \\ &= \frac{1}{|V|-1} \sum_{j \in V, j \neq i} P\left( d\left(Y_t^{(i)}, Y_0^{(i)}\right) < d\left(Y_t^{(i)}, Y_0^{(j)}\right) \right) \\ &= \frac{1}{V-1} \sum_{j \in V, j \neq i} 1 - \omega(i,j,t) \\ &= 1 - \bar{\omega}(i,t). \end{aligned}$$

**Temporal Dissonance for Embedding Diffusion**   Thus, we see that the continuous corruption simplifies to the mean probability of noisy embedding $Y_t^{(i)}$ being closer to some incorrect embedding $Y_0^{(j)}$ than the correct embedding $Y_0^{(i)}$. As with continuous signal degradation in the one-hot space, we do not have any dependency on the vocabulary size beyond averaging.

This reveals that embedding diffusion faces the same limitations as one-hot diffusion, regardless of whether tokens are obtained from noisy embeddings via dot product similarity or $l_2$ distance. We note that applying the insights of CANDI to embedding diffusion may work in practice. We choose to focus on one-hot diffusion due to the precise analytical control it offers over both discrete and continuous forms of corruption.

### B.5. Empirical Simulation for Embedding Diffusion

Here we provide empirical evidence to support the assumption of comparison independence used to derive the discrete identity corruption formula for embedding diffusion. To obtain the analytical expression for discrete identity corruption in the embedding space, we assume that the comparison events are independent: whether $Y_t^{(i)}$ is closer to $Y_0^{(j)}$ than $Y^{(i)}$ is independent across different incorrect tokens j.

While comparison independence provably holds for i.i.d. random embeddings, it is not obvious whether it holds for learned embeddings, which are structured and correlated. We validate this assumption empirically by comparing corruption curves for both random i.i.d. embeddings and learned GPT-2 embeddings (Figure 10). The close match between these two cases indicates that learned embeddings behave approximately the same as if comparison events were independent, justifying our modeling assumption.

**Experimental Setup**   Here we discuss the experimental setup for our empirical simulations. We choose to examine vocabulary sizes $V = \{5, 50, 500, 50, 257\}$. For each vocabulary size, we define a simulation round as selecting a token index $i$, retrieving the corresponding word embedding, adding noise and measure both discrete and continuous rank degradation per the formulas introduced in (18) and (27). For each vocabulary size and noise level, we perform 1,000 simulations.

**Simulating Smaller Vocabulary Sizes**   A crucial component of our analysis revolves around how discrete corruption scales with vocabulary size. We simulate smaller vocabulary sizes $|V|$ by randomly selecting $|V|$ embeddings to consider as the full embedding table. To ensure robustness, we repeat this random selection for 5 random subsets, except for $|V| = 50, 257$, where there is only one possible subset.

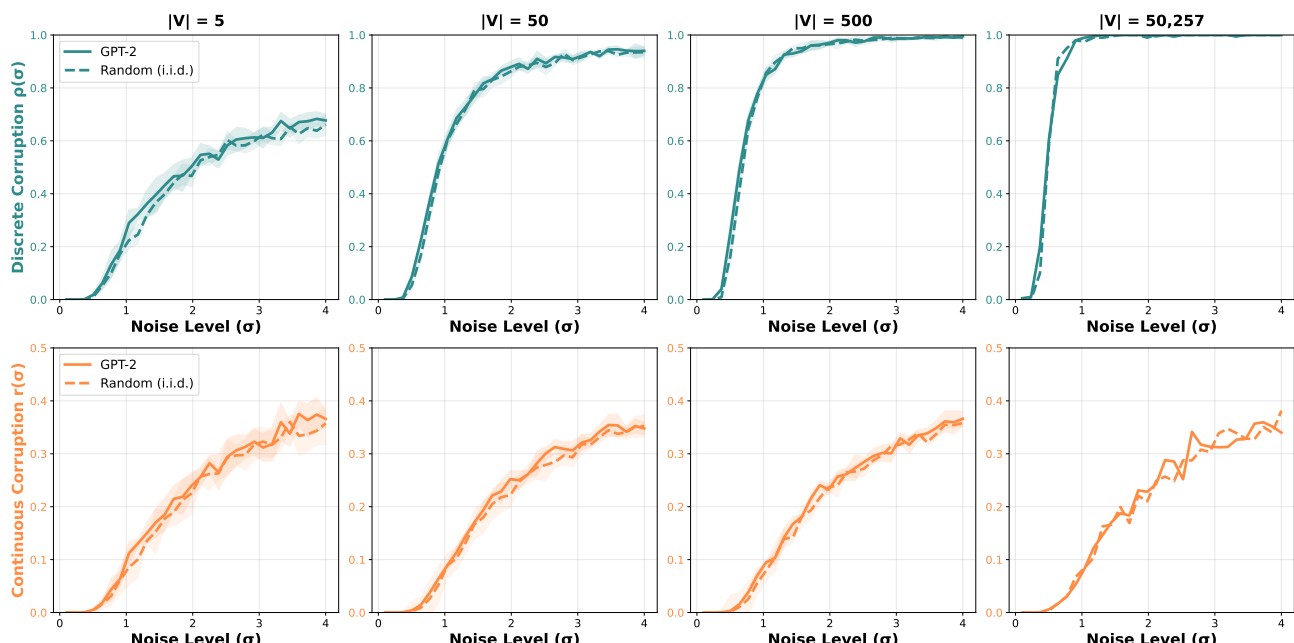

*Figure 10.* We observe that (1) temporal dissonance also occurs within the embedding space, for both learned and random embeddings; and (2) learned GPT-2 embeddings behave similarly to random i.i.d embeddings in terms of discrete and continuous corruption.

**Random Embedding Tables** We sample random embedding vectors by fitting an Isotropic Gaussian with the mean and variance of the learned embedding tables. This ensures that the sampled vectors are approximately the same scale as the learned embeddings, ensuring that the comparison of corruption rates is accurate.

**Results** We show the results in Figure 10, where we can make two important observations. First, **we note that the temporal dissonance phenomenon occurs with learned embeddings**: discrete corruption becomes more sensitive to small noise levels with larger vocabulary sizes whereas continuous rank degradation remains roughly the same. Second, we observe that the **GPT-2 embeddings exhibit the same corruption behavior as i.i.d random embeddings across vocabulary sizes**, validating our independence assumption as a practical modeling choice for analyzing temporal dissonance.

### B.6. Relation to Latent Diffusion

Here we discuss how our insights apply to latent diffusion methods. First we clarify the distinction between latent and embedding diffusion. We then discuss why latent diffusion avoids the temporal dissonance problem identified in this work. Finally, we discuss why latent diffusion still validates the core insight of our work and where the core difference lies.

**Latent Diffusion v.s Embedding Diffusion** Here we disambiguate between latent diffusion and embedding diffusion. Embedding diffusion refers to mapping each token to a continuous representation, and treating sequences of such representations as the diffusion space (Li et al., 2022; Dieleman et al., 2022; Gao et al., 2022). In latent diffusion, the continuous representation is obtained by applying an encoder to the sequence. Continuous diffusion models are trained within this latent space, and discrete text is obtained by using a trained decoder (Meshchaninov et al., 2025; Shabalin et al., 2025; Lovelace et al., 2023).

The key distinction is that in latent diffusion, each discrete position is a function of the entire continuous representation, as opposed to a one-to-one mapping between a continuous sequence and discrete sequence. More concretely, in embedding diffusion, a position $i$ in the embedding sequence depends only on token $i$. In latent diffusion, a position $i$ in the decoded sequence is a function of the entire latent sequence.

**Contextual representations and Temporal Dissonance** As token identities are distributed across the entire sequence as opposed to being constrained to their original position, Gaussian noising no longer induces a discrete corruption effect as the decoder can leverage the entire sequence to obtain information for a single position. Thus such methods naturally avoid

the implications of the temporal dissonance identified in this work.

**Latent Diffusion Leverages Discrete Corruption**   However, it is important to note that latent diffusion methods employ pretrained encoding and decoding components, which themselves are trained using discrete corruption through the masked language modeling objective (Meshchaninov et al., 2025; Shabalin et al., 2025; Lovelace et al., 2023). Thus latent diffusion methods validate the central insight of our work: discrete corruption is necessary to learn the conditional structure of language. The core distinction is *when* this discrete corruption occurs: in CANDI, we use hybrid corruption to simultaneously learn a continuous score function and discrete conditional structure. In latent diffusion, the encoder and decoder first learn to capture discrete conditional structure, and then a separate model is trained to learn a continuous score function.

## C. Architectural and Algorithmic Details

### C.1. Derivation of Structured Kernel Reverse Transition

Now we derive the reverse transition $P(X_s, M_s | X_t, M_t)$ that will allow us to reverse this structured noising process.

First, we note the following reverse transition probabilities for the masked vector $M_t$:

$$P(M_s[i] = 1 | M_t[i]) = \begin{cases} \frac{\alpha_s - \alpha_t}{1 - \alpha_t} & M_t[i] = 0, \\ 1 & M_t[i] = 1. \end{cases} \tag{28}$$

This directly follows from Shi et al. (2024); Sahoo et al. (2024), as this is simply the mask / unmasking probabilities from standard masked diffusion.

We start by the following decomposition:

$$P(X_s, M_s | X_t, M_t) = \sum_{M_s} P(M_s | M_t) P(X_s | X_t, M_t, M_s).$$

By definition of our forwards process, there are three possible transitions with non-zero probability: $P(M_s = 1 | M_t = 1)$, $P(M_s = 0 | M_t = 0)$, and $P(M_s = 1 | M_t = 0)$.

**Case 1:**   $M_s = 1, M_t = 1$. If $M_t = 1$, then $X_t = X_0$ and $X_s = X_t$ by definition of the noising kernel. Thus $P(X_s | X_t, M_t = 1, M_s = 1) = \delta(X_s - X_t)$.

**Case 2:**   $M_s = 0, M_t = 0$. In this case, both $X_s, X_t$ remain noisy one-hots $\tilde{X}_s, \tilde{X}_t$ and we have that the distribution over $X_s = \tilde{X}_s$ is $P_{\text{Gaussian}}(X_s | X_t)$, which is defined by the continuous diffusion process.

**Case 3:**   $M_s = 1, M_t = 0$. In this case, we have that $M_s$ transitions to being clean. We can approach this by marginalizing over $X_0$:

$$\begin{aligned} P(X_s | X_t, M_t = 0, M_s = 1) &= \sum_{X_0} P(X_0 | X_t) P(X_s | X_t, X_0, M_t = 0, M_s = 1) \\ &= \sum_{X_0} P(X_0 | X_t) \delta(X_s - X_0) \\ &= P(X_0 = X_s | X_t). \end{aligned}$$

Thus, we arrive at the following reverse transition:

$$P(X_s, M_s | X_t, M_t) = \begin{cases} \delta(X_s - X_t) \cdot \delta(M_s - 1), & M_t = 1 \\ P(X_0 = X_s | X_t) \cdot \delta(M_s - 1), & M_t = 0, \text{ with prob } \frac{\alpha(s) - \alpha(t)}{1 - \alpha(t)} \\ P_{\text{Gaussian}}(\tilde{X}_s = X_s | \tilde{X}_t) \cdot \delta(M_s - 0), & M_t = 0, \text{ with prob } \frac{1 - \alpha(s)}{1 - \alpha(t)} \end{cases}$$

## C.2. Full Exact Algorithm

Here we provide details for the exact inference algorithm for our method. For a given state at time t, we have the noisy vector $X_t$, and the mask $M_t$ which indicates which positions are corrupted with $M_t[i] = 1$ and which positions are uncorrupted with $M_t[i] = 0$. For a target time $s$, we first sample which positions will transition from corrupted to uncorrupted using the following:

$$M'_s \sim \text{Bernoulli}\left(\frac{\alpha(s) - \alpha(t)}{1 - \alpha(t)}\right).$$

We label this as $M'_s$ as we must carry-over the previous clean positions from $M_t$ to obtain $M_s$. For positions where $M'_s[i] = 1$, we sample tokens $X'_s$ from $P_\theta(\cdot|\tilde{X}_t)$ and convert it to a one-hot $\tilde{X}'_s$. For positions where $M'_s[i] = 0$, we apply the update formula for the reverse probability flow ODE:

$$\tilde{X}''_s = \tilde{X}_t - \frac{1}{2}(\sigma(t)^2 - \sigma(s)^2) \cdot \nabla \log p(\tilde{X}_t). \tag{29}$$

The score function $\nabla \log p(\tilde{X}_t)$ is estimated using $P_\theta(\cdot \mid \tilde{X}_t)$ via (4). To obtain the final state $\tilde{X}_s$, we must carry over the clean positions where $M_t = 1$; update the new clean positions where $M_t[i] = 0, M'_s[i] = 1$, and update the noisy positions where $M_t[i] = M'_s[i] = 0$. The final update rule is given as follows:

$$\tilde{X}_s = M_t \odot \tilde{X}_t + \neg M_t \odot M'_s \odot \tilde{X}'_s + \neg M_t \odot \neg M'_s \odot \tilde{X}''_s. \tag{30}$$

This enables both discrete conditioning and continuous joint evolution by using the output of the model to serve as $\mathbb{E}[\tilde{X}_0|\tilde{X}_t]$. However, this comes at the cost of having to perform matrix multiplication of this expectation against the model embedding table. We include the full pseudo-code in Algorithm 1.

---

**Algorithm 1** Exact Joint Inference

---

**Require:** trained model $P_\theta$, time-steps $S = \{t_i\}, 1 = t_1, t_2 \ldots t_n \approx 0$
1:   $X_t \sim \mathcal{N}(0, \sigma(t_1)I)$
2:   $M_t = [0, 0, \ldots 0]$
3:   **for** $t, s$ in $S[: -1], S[1 :]$ **do**
4:      $\hat{X}_s \sim P_\theta(X_0|X_t)$
5:      Sample $u \sim U[0, 1]$, Set $M'_s = \mathbf{1}\left[u < \frac{\alpha(s) - \alpha(t)}{1 - \alpha(t)}\right]$
6:      Calculate $\nabla \log p_t(X_t) = -\frac{X_t - \mathbb{E}[X_0|X_t]}{\sigma(t)^2}$ using $P_\theta(X_0|X_t)$
7:      Compute Euler-Maruyama ODE update as $\tilde{X}_s = X_t - \frac{1}{2}(\sigma(s)^2 - \sigma(t)^2)\nabla \log p_t(X_t)$
8:      $X_s \leftarrow X_t \odot M_t + \neg M_t \odot M'_s \odot \hat{X}_s + \neg M_t \odot \neg M'_s \odot \tilde{X}_s$
9:      $X_t, M_t \leftarrow X_s, (M_t \vee M'_s)$
10: **end for**
11: return $\text{argmax}(X_t)$

---

## C.3. Details on Sampling Trick

Here we describe our approximate inferece algorithm, which we use to simulate the reverse trajectory without computing the costly matrix multiplication required.

While our model is trained on noisy one-hots, the transformer component of the model only sees $Y_t = W X_t$, where $W$ is the input embedding look-up table learned by the model. For a given trajectory $X_0, X_{t_1}, \ldots X_{t_n}, X_T$, we have a trajectory in the embedding space $Y_0, Y_{t_1}, \ldots Y_T$. This corresponds to the trajectory the transformer actually sees throughout sampling.

In the exact inference algorithm, the transitions from $Y_t$ to $Y_s$ is computed by first obtaining $P(X_0|X_t)$, computing the ODE update to $X_t$ to obtain $X_s$, and then performing matrix multiplication to obtain $Y_s$. We can write this as follows and

use the linearity of matrix multiplication:

$$
\begin{aligned}
Y_s &= W \left( X_t - \frac{1}{2}(\sigma(t)^2 - \sigma(s)^2) \nabla \log p_t(X_t) \right) \\
&= Y_t - \frac{1}{2}(\sigma(t)^2 - \sigma(s)^2) W \left( \frac{X_t - E[X_0|Y_t]}{\sigma(t)^2} \right) \\
&= Y_t - \frac{1}{2}(\sigma(t)^2 - \sigma(s)^2) \left( \frac{Y_t - E[Y_0|Y_t]}{\sigma(t)^2} \right).
\end{aligned}
$$

Thus, given some initial $Y_y$, the matrix multiplication is only mathematically necessary compute $\mathbb{E}[Y_0|Y_t] = W P_\theta(X_0|X_t)^T$. While this matrix multiplication is expensive, we note it is quite cheap to sample token indices from $X_0 \sim P_\theta(\cdot|X_t)$ and use this this to look up $Y_0$. By monte-carlo sampling $Y_0$ in this manner and averaging them, it is possible to approximate $\mathbb{E}[Y_0|Y_t]$. Thus, we propose to sample from $P_\theta(X_0|X_t)$ and perform token look-up instead of computing the full matrix multiplication. We find that using a single sampling step is sufficient to obtain good generation quality, thus making the cost of the inference algorithm similar to that of pure discrete methods that only operate in the token space.

It should be noted that this is the same single-step Monte Carlo approximation used in Cetin et al. (2025). However, while they apply this specifically for embedding diffusion, we demonstrate it is also suitable for one-hot diffusion. We include the full pseudo-code in Algorithm 2.

---

**Algorithm 2** Approximate Joint Inference

---

**Require:** trained model $P_\theta$, time-steps $S = \{t_i\}, 1 = t_1, t_2 \ldots t_n \approx 0$
1: $\tilde{X}_t \sim \mathcal{N}(0, \sigma(t_1)I)$
2: $Y_{\text{cache}} \leftarrow W_{\text{embedding}} \tilde{X}_t \triangleright$ *Cache initial noisy embeddings*
3: $X_t \leftarrow \text{argmax}(\tilde{X}_t) \triangleright$ *Get discrete tokens via argmax*
4: $M_t = [0, 0, \ldots 0]$
5: **for** $t, s$ in $S[: -1], S[1 :]$ **do**
6: $\quad Y_t = (W[X_t]) \odot M_t + Y_{\text{cache}} \odot \neg M_t \triangleright$ *Use tokens lookup instead of matrix multiplication*
7: $\quad X'_s \sim P_\theta(\cdot|Y_t) \triangleright$ *Sample tokens $X'_s$ from learned posterior*
8: $\quad$ Sample $u \sim U[0, 1]$, Set $M'_s = \mathbf{1} \left[ u < \frac{\alpha(s) - \alpha(t)}{1 - \alpha(t)} \right]$
9: $\quad \nabla \log p_t(Y_t) = -\frac{Y_t - \mathbb{E}[Y_0|Y_t]}{\sigma(t)^2}$ with $\mathbb{E}[Y_0|Y_t] \approx W\hat{X}_s \triangleright$ *Use the sampled tokens for single step monte-carlo approximation*
10: $\quad$ Compute ODE update as $Y_{\text{cache}} = Y_t - \frac{1}{2}(\sigma(t)^2 - \sigma(s)^2)\nabla \log p_t(Y_t)$
11: $\quad X_s \leftarrow X_t \odot M_t + \neg M_t \odot M'_s \odot X'_s$
12: $\quad X_t, M_t \leftarrow X_s, (M_t \vee M'_s)$
13: **end for**
14: return $X_t$

---

### C.4. Details on Classifier-based Guidance

Here we discuss how to leverage gradient information to guide inference towards desired sample properties. Let $f_\phi : \mathcal{X} \to C$ be a classifier over class labels $C$. Given a latent sample $X_t$ from the hybrid diffusion trajectory, we apply the following mapping to represent it in a form that allows gradients:

$$
\hat{X}_t = \text{Onehot}(\text{argmax}(X_t)).
$$

During inference, we simply add a scaled version of this gradient to the continuous component of the reverse step update. Specifically, we compute the ODE integration step as follows:

$$
\tilde{X}''_s = \tilde{X}_t - \frac{1}{2}(\sigma(t)^2 - \sigma(s)^2) \cdot \left( \nabla_{\tilde{X}_t} \log p(\tilde{X}_t) + w \cdot \nabla_{\hat{X}_t} f_\phi(\hat{X}_t) \right). \tag{31}
$$

This demonstrates a key advantage of the hybrid approach: pure discrete diffusion methods must train bespoke classifiers tailored to the specific corruption pattern and carefully determine how to translate gradient information into posterior steps. In contrast, our method is fully compatible with classifiers trained without diffusion-specific modifications, and only requires gradient addition.

### C.5. Parameterization and Training Details

**Training Objective**    Given that our model is learning the posterior $P_\theta(\cdot|X_t)$, we use the following objective:

$$\mathcal{L}(\theta, D) = \mathop{\mathbb{E}}_{\substack{X_0 \sim \mathcal{D}, \, t \sim [0,1] \\ X_t \sim q(\cdot|X_0)}} \left[ -\frac{1}{1 - \alpha(t)} \log P_\theta(X_0|X_t) \right]. \tag{32}$$

This objective serves as a suitable optimization target as it uses the number of corrupt positions, $1 - \alpha(t)$, to balance the prediction loss across all time-steps. This ensures that each corrupt position recieves equal weighting, regardless of the gloval time $t$.

Furthermore, this corresponds to the ELBO for standard masked diffusion, differing by the forward noising kernel. This demonstrates that hybrid diffusion can be achieved by simply re-purposing the existing framework from masked diffusion.

**Corruption Bias**    One crucial component of the hybrid denoising process is the masked variable $M_t$, which indicates which positions in $X_t$ are corrupted and which positions are clean. We implement this component of the schedule using a learnable bias vector $b \in \mathbb{R}^d$, where $d$ is the embedding dimension for the model. Specifically, if the position $i$ has $M_t[i] = 0$, we calculate the embedding as:

$$e_i = (1 - \lambda) \cdot W\tilde{X}_t[i] + \lambda \cdot b$$

Clean positions where $M_t[j] = 1$ are calculated as $e_j = W\tilde{X}_t[j]$. In practice, we set $\lambda = .5$. This bias ensures that the architecture reflects the theoretical definition of the denoising process, which requires some kind of variable to indicate which positions are corrupted or clean.

Furthermore, this choice of implementation allows for the **unification between this hybrid and masked denoising processes**. When $\lambda = 1$, all corrupted positions receive the same bias embedding, recovering the behavior of masked diffusion.

**Preconditioning**    Following the work of Karras et al. (2022), we use pre-conditioning to ensure that the model inputs are roughly the same magnitude across different time regions. Our preconditioning is designed to ensure that the clean positions maintain the same magnitude while the noisy positions are rescaled to ensure the magnitude of inputs remain the same across time-steps. We define the preconditioning function $f$ given embeddings $Y_t$, mask $M_t$, and continuous noise value $\sigma(t)$ as follows:

$$f(Y_t, M_t) = \neg M_t \odot \frac{Y_t}{\sqrt{\sigma(t)^2 + 1}} + M_t \odot Y_t.$$

By leveraging the mask vector $M_t$, our preconditioning function ensures that the magnitudes for the clean positions remain the same, whereas the noisy positions are rescaled to ensure stable magnitudes across timesteps.

## D. Experimental Results

### D.1. Frontier Analysis

Here we explain why we choose frontier analysis as our primary means of comparing different methods, and why we believe such analysis is necessary. First we describe why we focus on generative quality as opposed to likelihood evaluations. Next we discuss why we leverage frontier analysis as opposed to using a fixed temperature for all methods. Finally we explain why frontier analysis is necessary for fair comparisons.

**Generative Quality v.s Likelihood Evaluation**    Throughout our experiments, we focus on measuring generative quality as opposed to likelihood evaluations. We avoid likelihood-based metrics as they assume infinitesimal steps ($dt \to 0$), whereas

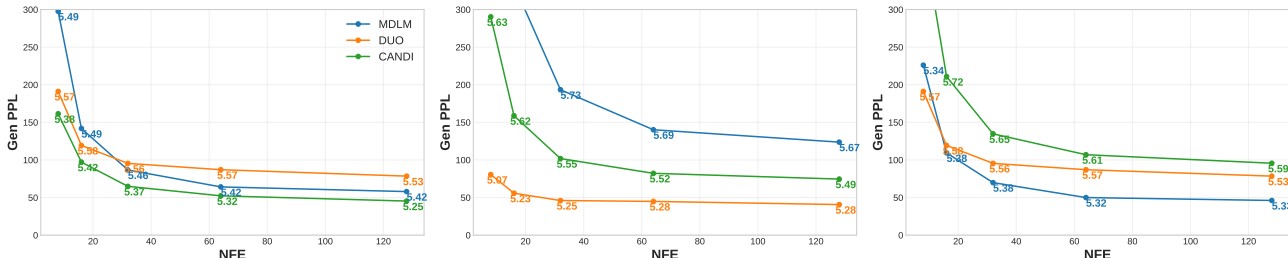

*Figure 11.* We demonstrate that single-point evaluations for perplexity and entropy can lead to misleading results. For each plot, we adjust the temperature of each method ($\tau \in [.875, 1.0]$) to achieve reasonable ranges of entropy (5.2-5.7, annotated along the curves) (Sahoo et al., 2025; Zheng et al., 2024). We show that small changes in this hyperparameter can produce entirely different relative performance rankings. We provide details on temperature settings for each plot in Appendix D.1.

our focus is practical generation with a finite number of timesteps. Moreover, ELBO comparisons are inherently unfair for hybrid methods. Considering only the discrete component favors CANDI, since its continuous component carries more information per timestep than mask tokens. Including both discrete and continuous components, however, compares likelihoods over fundamentally different spaces, making the comparison invalid.

**Frontier Analysis** To compare generative quality across models, we extend the *entropy-perplexity frontier analysis*, which captures the trade-off between sample diversity (entropy) and fluency (perplexity) (Zheng et al., 2024). For each method, we vary the temperature used for sampling tokens, which results in different operating points of diversity v.s coherence. We compare methods by plotting the entire frontier for each method. Frontiers are essential because each model defines an entire operating ranges in terms of diversity and coherence, and thus focusing on a single fixed temperature may not capture the full extent of a methods capabilities.

Furthermore, using a fixed temperature for all methods may not reflect the best performance for each method, as different methods may have different optimal temperature settings. We motivate the usage of different temperatures by noting that, within autoregressive text evaluation, different temperatures are often used for the same model across different tasks (Achiam et al., 2023; Touvron et al., 2023). If a single model's optimal temperature depends on the task, different generative models are likely to have different optimal settings. As a result, using a global temperature for all methods does not reflect true practical performance.

**The Single-Temperature Trap.** Figure 11 illustrates why single-temperature evaluations can produce misleading comparisons. We evaluate MDLM (Sahoo et al., 2024), DUO (Sahoo et al., 2025), and CANDI at NFE $\in \{8, 16, 32, 64\}$ on OpenWebText (Gokaslan et al., 2019), selecting temperatures that produce entropy in the standard range (5.0–5.6) used by prior work (Zheng et al., 2024). Notably, constraining entropy to similar ranges (e.g., all methods at entropy $[5.25, 5.7]$) is insufficient—small temperature variations within this range still flip rankings. Since temperature tuning is necessary for fair comparison, the only robust solution is to plot complete frontiers by sweeping through temperature values. A method genuinely outperforms if its frontier dominates at all entropy levels, ensuring the conclusion holds across all reasonable temperature settings.

To obtain the perplexity v.s NFE curves from Figure 11, we use the following temperature settings:

- Left Plot: MDLM $\tau = .925$, DUO $\tau = 1.0$, CANDI $\tau = .875$

- Middle Plot: MDLM $\tau = 1.0$, Duo $\tau = .9$, CANDI $\tau = .925$

- Right Plot: MDLM $\tau = .9$, Duo $\tau = 1.0$, CANDI $\tau = .95$

By using temperatures within the narrow range of $.875 - 1.0$ for all methods, it is possible to achieve completely different relative rankings without changing the underlying models. This is because each model defines an entire region of operating points in terms of diversity and coherence. As a result, single point temperature comparisons only evaluate specific points within this region, which do not faithfully reflect the entire scope of the model's generative capabilities.

## D.2. Vocabulary Scaling Experimental Details

**Model Architecture**    We use a diffusion transformer from Peebles & Xie (2023) with a hidden dimension of 768, 12 transformer blocks, and a context length of 1024. We do not tie the embedding tables to the language model head.

**Embedding Diffusion Model**    While the pure continuous one-hot model follows the same SDE as defined for CANDI, this is not possible for the embedding diffusion model, as the embedding space does not have a closed-form solution for continuous rank degradation. To maintain similarity with the one-hot diffusion model, we apply a VE SDE from Song & Ermon (2019). We use the following noise schedule for both training and inference:

$$\sigma(t) = \sigma_{\min} \left( \frac{\sigma_{\max}}{\sigma_{\min}} \right)^{2t}.$$

We set $\sigma_{\min} = .01, \sigma_{\max} = 2.0$ across both Text8 and OWT. We sample $t$ uniformly, using the low variance sampler described in Sahoo et al. (2025). We train the entire model, including the embeddings, using an unweighted cross entropy loss, as this has been shown to avoid degenerate embedding problems (Dieleman et al., 2022).

**On Pretrained Embeddings**    One strategy for embedding diffusion is to leverage pretrained embeddings as a static encoding for continuous diffusion. However, we do not examine such methods as pretrained embeddings are not a valid test of our hypothesis. Our goal is to understand whether continuous diffusion can learn discrete conditional structure. Pretrained embeddings (from autoregressive or masked diffusion models) provide this structure as an external input. If pretrained embeddings succeed, this only confirms that continuous diffusion can denoise in well-structured spaces, which is orthogonal to our research question. If they fail, the failure is ambiguous and could reflect embedding-diffusion incompatibility rather than the fundamental limitation we investigate.

We therefore test end-to-end learning, which directly examines whether continuous diffusion can simultaneously learn both discrete dependencies and continuous geometry. This maintains fair comparison with discrete diffusion methods (MDLM, DUO), which also learn all structure from scratch.

**Training Details**    We keep training hyper-parameters the same for both Text8 and OWT. On Text8, we train for approximately 1,000 steps using a batch size of 512, a learning rate of $3e - 4$ and 250 warm-up steps. We use the AdamW optimizer (Loshchilov & Hutter, 2017) with a learning rate of 1e-3. We also keep an EMA copy of the model weights with the decay set to 0.9999.

For all experiments, we use 8 Nvidia H100s. The training runs for Text8 complete in roughly 20 minutes, and the training runs for OWT take a little over 12 hours. For each dataset, all models are subjected to roughly the same number of training steps, ensuring any underperformance is not due to undertraining.

**Sampling Details**    For Text8, we sweep temperatures from .1 to 1.3 in intervals of .1 to produce the frontier graphs. For the OWT results in Figure 6, we sweep through 20 evenly-spaced temperatures between .1 and 2.0. We generate 128 batches for each temperature point. We use float64 precision for sampling, as lower precisions has been shown to be equivalent to implicit temperature tuning (Zheng et al., 2024). All sampling experiments were done on a single NVIDIA RTX A6000.

**Text8 Metrics**    Text8 is a character level dataset, so we count the number of unique words in the dataset by delimiting the text using white-spaces and setting all the characters to be lowercase. We apply a similar methodology to count words within generated sequences. In order for a word in a generated sequence to count as valid, it must correspond to some word that occurs in either the training or test dataset.

## D.3. OWT Text Generation

We describe the methodology and additional results for the text generation results. Note the OWT results from Figure 6 use partially trained checkpoints, whereas the results from Figure 7 reflect fully trained model ability.

**Training**    We train CANDI for around 1,000,000 training steps on OWT, on OpenWebText with sequence packing and GPT-2 tokenizer (Gokaslan et al., 2019; Radford et al., 2019). Following prior work, we use the diffusion transformer from Peebles & Xie (2023) with 110 million parameters and train for 1,000,000 steps (Arriola et al., 2025; Sahoo et al., 2025;

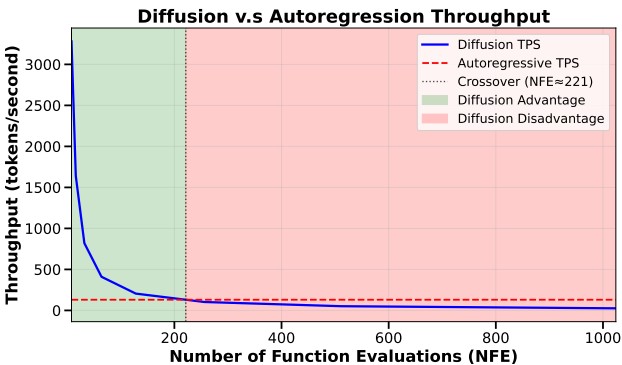

*Figure 12.* We show the throughput for autoregressive models and discrete diffusion models, using GPT-2 as a representative autoregressive model and MDLM as a representative diffusion model. Due to the advantage of native key-value caching, autoregressive methods have a speed advantage over diffusion models once the NFE crosses $\approx 200$.

2024; 2025). For the DUO and MDLM baselines, we use the official checkpoints provided by Sahoo et al. (2025; 2024). We use 8 H100 GPUs, and find that training takes around 7 days.

**Temperature Sweep Details**    For the OWT results in Figure 7, we sample using each method for the set NFE and vary temperature $\tau$ within the range of $[.7, 1.0]$ with increments of $.025$. We choose to use a different range of temperatures between Figure 6 and Figure 7 as each emphasizes different aspects: in Figure 6, we are interested in demonstrating the full extent of mode failure for continuous diffusion; and in Figure 7, we want to demonstrate the benefits of CANDI within a practical operating range of entropy-perplexity. As before, we collect 128 samples for each method and use the mean entropy and perplexity to construct the curves.

**Through-put Advantage of Discrete Diffusion over AR**    To compute the throughput advantage of discrete diffusion over autoregressive generation, we use masked diffusion as a representative diffusion model, since DUO, Candi approximate, and MDLM inference have the same algorithmic steps. We generate 20 sequences of length 1024 using GPT-2 using the HuggingFace transformers package, and we run masked diffusion for varying NFE for the same number of sequences and sequence length. We include the frontier plot we use to obtain the throughput ratios below.

**Frontier Curves for Higher NFE**    Here we include additional frontier curves for algorithms at higher NFE in Figure 13. We observe that CANDI does slightly underperform at higher NFE. This aligns with theoretical expectations: at higher NFE, less tokens need to be updated simultaneously, decreasing the benefits of gradient information. Interestingly, we also

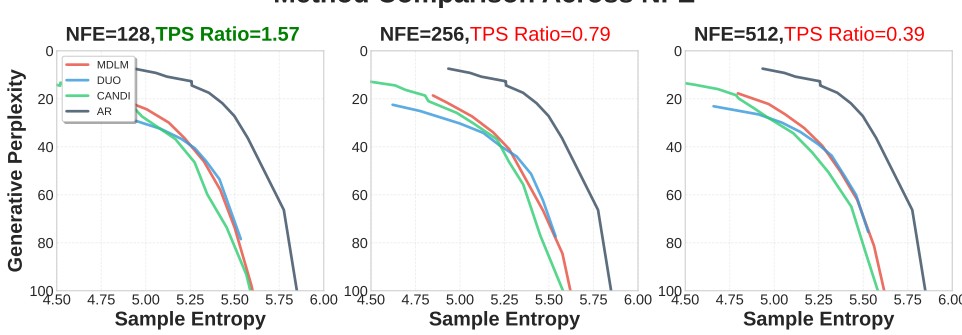

*Figure 13.* While not achieving the best frontier, CANDI maintains competitive performance with strong pure-discrete baselines. As the number of NFE decreases, discrete conditional information becomes more important as fewer positions need to be updated at the same time.

observe a general trend that discrete diffusion models do not benefit from increased NFE in terms of frontier expansion – in Figure 14, we show the overlapping frontiers for each method by NFE. We observe that the frontiers are largely the same,

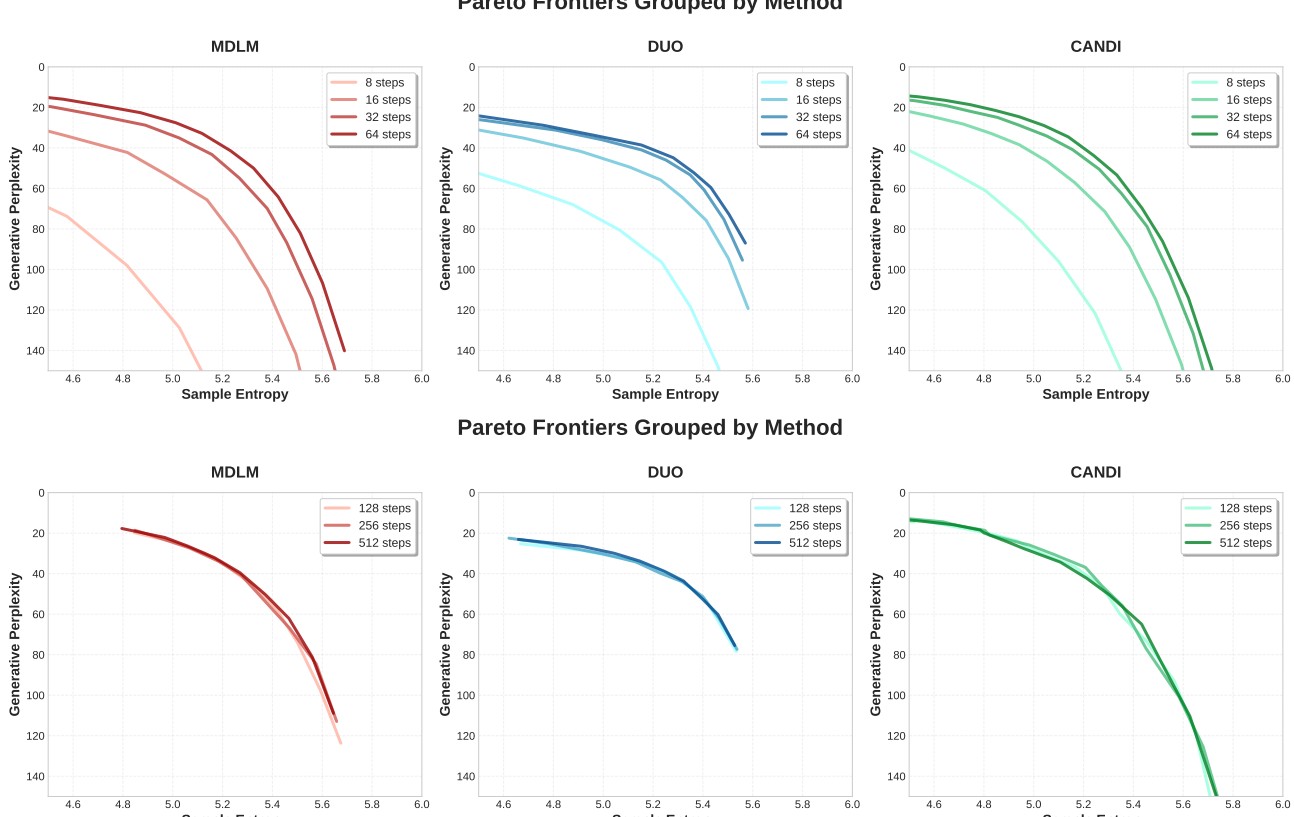

*Figure 14.* We show achievable frontier for MDLM, DUO, and our method for both small and large NFE. A thorough temperature sweep reveals that more NFE does not necessarily lead to significant improvements in terms of the actual scope of generative ability: past 128 NFE, all selected methods do not significantly improve with increased NFE. The effect of more steps can be achieved with a simple grid search, at least when using perplexity as a measure of generative quality.

implying that the performance improvements obtained by increasing NFE can be achieved via temperature tuning. We do not know whether this is simply an artifact of this specific scale of models, or whether this is a persistent phenomenon for larger diffusion models.

**Comparison Between Approximate and Exact Inference Algorithm** We also compare our approximate algorithm with the exact variant of our algorithm and discover that the approximation does not degrade the generative quality as represented through the entropy and perplexity frontier. This aligns with observations from Cetin et al. (2025), where they find the approximation of the expectation via single-step Monte Carlo entropy.

**Generated Samples** We include generated samples in Appendix D.5.

### D.4. QM9 Guidance Details

**Model Architecture** For both diffusion based models and all classifiers, we use the diffusion transformer architecture. For the diffusion base models, we use a diffusion transformer with a hiddden dimension of 768, 12 layers, and 12 attention heads per layer. We use a dropout rate of .1.

For the classifiers, we use a hidden dimension of 512, 8 transformer blocks with 8 attention heads each. We use the standard settings from the codebase provided by Schiff et al. (2024).

**Training Details** For the base diffusion model, we train for 10,000 steps with a batch size of 512 and a learning rate of $3e-4$. Schiff et al. (2024). We use this to train UDLM, MDLM, and CANDI.

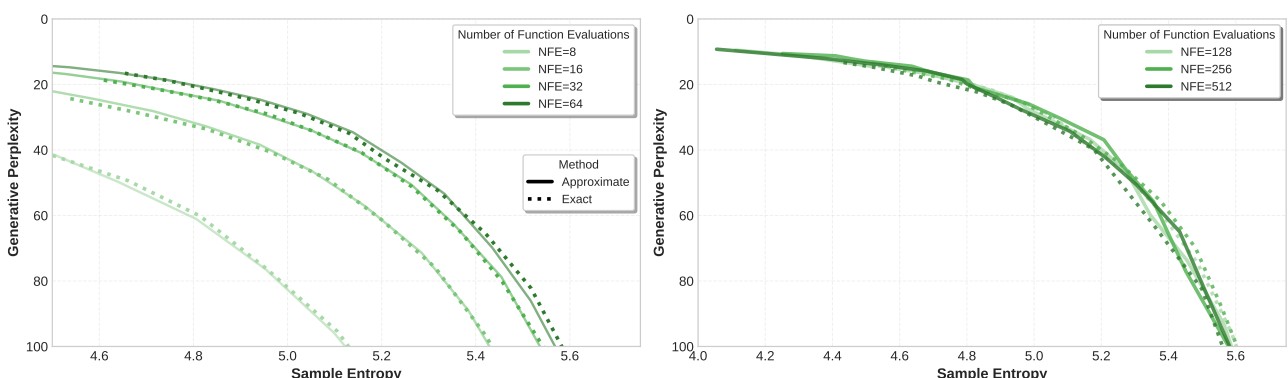

*Figure 15.* We show that despite using a single step Monte Carlo estimate of the expectation, the approximate algorithm is competitive with the exact algorithm, and in some cases slightly better. We also note that CANDI does not improve past a certain number of NFE steps – this is due to the fact that CANDI learns continuous gradient information, which is only useful when multiple tokens per step are being updated. As NFE increases, less positions need to be updated simultaneously.

For the diffusion classifiers, we train 10,000 steps with a batch-size of 512 for the same number of steps. For UDLM and MDLM, we train a classifier for both diffusion methods that learns to classify inputs corrupted using their respective corruption kernels. For CANDI, we train a classifier without any input corruption to match the settings for a generic, off-the-shelf external classifier.

**Sampling Details** For sampling, we evaluate all methods at NFE 16 and 32 for a range of guidance strengths and sampling temperatures. We show the best frontiers for each method, which obtain by running a grid search for each method at each NFE. We stop when temperature modifications cease to result in strictly dominating frontiers for each method. We include the full temperature sweep in Figure 16.

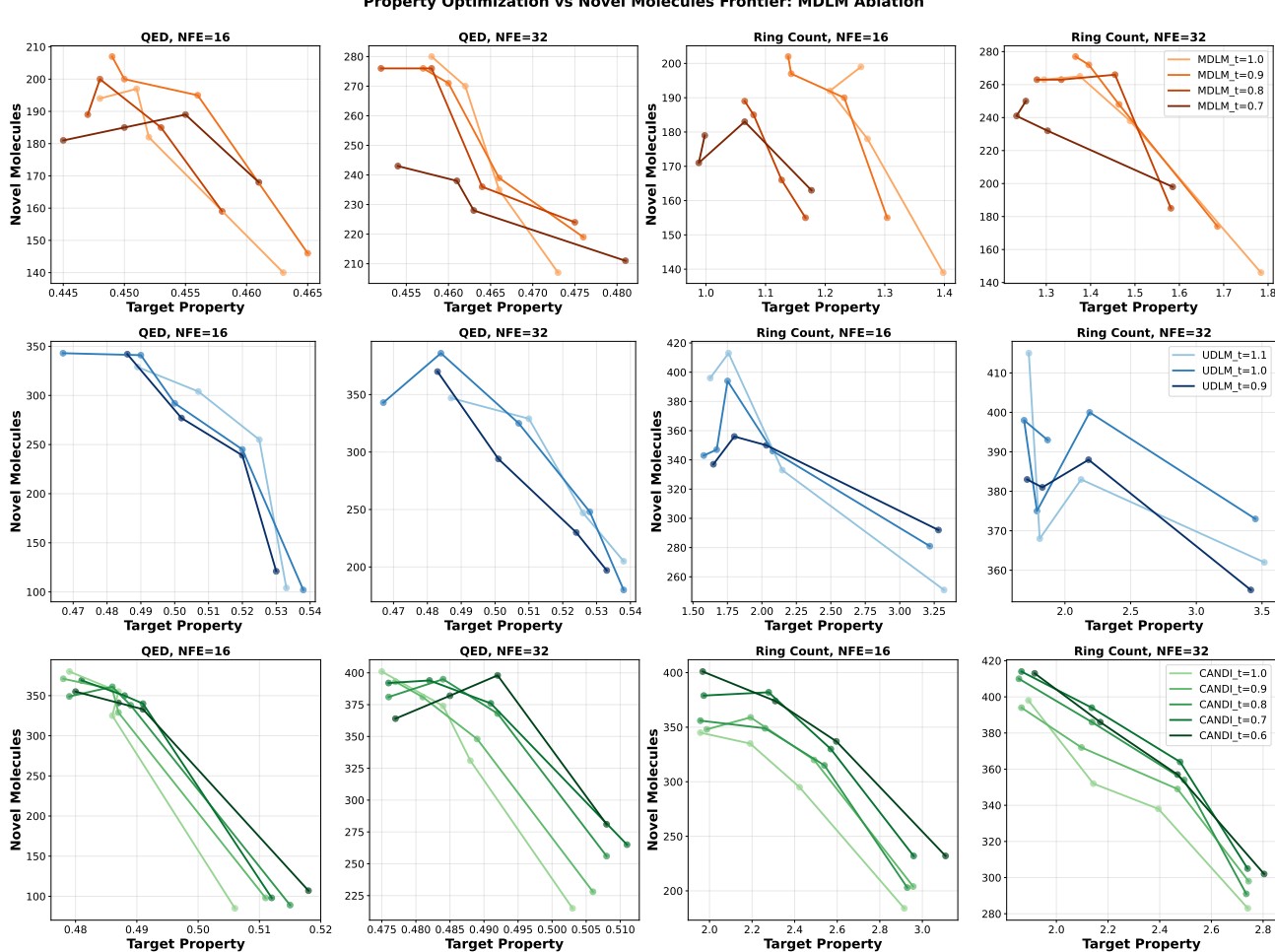

*Figure 16.* We include the different frontiers for each method obtained by applying grid search to inference-time temperature values. We observe that temperature $\tau = .9$ roughly achieves the best frontier for MDLM; $\tau = 1.0$ achieves the best frontier for UDLM, and $\tau = .7$ achieves the best frontier for CANDI across both properties and NFEs. While some frontiers may overlap, this would not significantly impact relative comparisons between UDLM, MDLM, and CANDI.

## D.5. Generated Samples

¡—endoftext—¿. population at 3,546 per cent of GDP.
I hope at the same time as, for Americans, the two take advantage of this opportunity to represent nations. Because the U.S. is probably the most varied source of news, and theirs is to feel that it has international power, and easy- to-see. With Americans, it will certainly get a lot of press. I feel the U.S. is one of the world's strongest powerful, and really takes the world's media world.
And so the opposite. For decades, the world has been focused in the U.S. for the benefit of world news coverage with visits globally.
At the United Nations Summit of the Americas in 2010, "Robert Gibbs," the spokesman for the U.S. Congress, stated, "Why the world is an this place for the truth is what the American press, really is."
Naturally, it was time for the American people to meet U.S. and the media for the first time. But, from the the U.S. perspective, it's something more important. The U.S. president even flies on a last flight to provide people with the news at the summit of more than 500 out of the 500 who had attended and had been there before.
Americans dominate online and other news from live, online TV
With the past, the Internet is the main source of news outside of world. The economics are clear, and people who are interested in the thefind a website that can be designed to maximize the effort and effort they spend. New York's websitesee what I haven't is that, even in its final days, the U.S has a substantial presence in global social media news, the same size as Japan. In 90S, Japan has the top 5
But it isn.t the only way to be. There has been a lot of apparently biased reporting from journalists about shows featuring Democratic candidates that won't be broadcast, or this is mostly about U.S. politics, and in the U.S., you can actually see the U.S. is crucial.
What the wrong with the U.S.
To top it off, politicians, the 28-year-old president in the U.S., aren't necessarily as much undervalued as journalists, and in most Americans.
Press is not about what it should be covered for and there's no way to tell
While long-term policy and the impact of international relations may have been limited, the press is also about the entire world market. When the United States shows there are no problems for the likes of China and Japan, and makes a "reminder check, you" needs a certain number of politicians willing to sell their "agricably-dependentvelvet" prime policy with the same countries as the few "non-rich" countries which would get coverage in such a way. (Which brings you to most cases of the United States getting so covered on world politics).
The United Kingdom. also works in a way besides the United States as is the case. The EU wants to deal with China, but it't have to deal with the UK for them to continue with or accept the North American Free Trade Agreement, under which President Barack Obama has very very carefully embarcs. The only the things that can help us keep the U.S. trade market the way it would is that Britain pays know or care more about its multi-billion-dollar holdings while the country approves of free trade unions. The average American doesn't necessarily have much more detailed coverage that would disappear from most of us, her children, or one of our grandchildren that is cleaning up from the world's pollution.
Unlike in Europe, certain countries such as the U.S. on the WTO don't need permission from the rest of the world to enter the WTO, as Obama last night has expected to ask. Xi Jinping has said there are no "sanffi" because what Chinese media must do is what they land on anywhere in other countries. The U. country, which is the negotiating party to the World Trade Council, proudly claim that it has no real concerns about its market's status, and has the most aspect of that other than freedom of trade in the trading area.
China's press coverage has really more things to do, according to the most recent US from conducted by Trade Un, the website of a consortium of Belgian, French, and Dutch, data points, in which there were 63 Chinese. journalists, the most stories that were not related to the United States. In some instances, some media, such as The Dragon, former Prime Minister Bloomberg, has visited China's court.
¡—endoftext—¿

*Figure 17.* Sample Generation from CANDI at NFE=16 with temperature $\tau = .85$. At this NFE, CANDI generations attained an average generative perplexity of $71.43$ with an entropy of $5.29$.

¡—endoftext—¿ It hasn't been a resolution yet, but it is believed to be the starting point of talks.
The opposition votes include Israel de facto France. The northern France abstained alongside the Mauritius and Luxembourg countries in the 7,371 vote, which was held in opposition.
Fate Florian Refouches said the strong vote showed unity.
"It will only be a pand miracle, a handshake and a gun of the population to see peace in Israel," Refouches after the vote. "Christians want to honored together and everybody wants the road of globalization blocked and want peace always to continue, for the full strength of the people."
The French voted for peace, but a lot of Israel's citizens needed to be with them. Hope was fading.
"They sought ebola, but nothing came," Lefré, a businessman in Roxanne France. Father N'Nab, which blames Israel for California's outbreak, added.
Bigger a holiday began as the 600th anniversary of a special holiday is established just before the vote.
One of the most ambitious projects of people like Manuel Martiel, a member of the campaign pro-Israel, 'TALLISINJA: A Manifesto for Palestine Liberation', promised ' to steal 20,000 hostages taken from in the Arab world by the Israelites'.
Abdelene and his wife now continue their support for Roxanne.
"My all-consuming love will be a great direct contribution to Israel," he told the Le Monde journalist.
"It should be a message to all," Abdelene says. "I am more concerned against more destruction than that of Les Mot d'Arc. More."
On the main streets of the state, the voters give a third of the choice to Israel, to those within the party. Support stands at 74 percent.
You can see the face-to-face political coverage from political parties on Monday.¡—endoftext—¿Eight elected officials from across the country are using their pastimes state powers to allow politicians to apply money to settle differences between workers and their concerns. The idea on that board is meant for the state to deliver shovel-ready legislative revenue to a major passenger.
Represents by a Democratic lawmaker.
When Governor Cuomo told the AFL-CIO's reporters in the same piece discussing the fallout from trying to target Democrats with the problems that matter to his constituents, " I'm a politically neutral legislator."
Reservative.
According to Cuomo (who has never been a graduate of high school before and was previously a public school teacher in New York), he was a student and professional prodigy. He loved his "down the earth" commute. "My family don't like commuting. I love it to be miserable. I love to be through traffic wherever I want to."
The governor has plenty of time remaining for opportunity to negotiate with the commuter rail companies. He's hoping for them to get that through legislation due to be delivered in late May.
The state, most people could not be open to lowering the minimum wage. That is why this is an intense primary issue. The upcoming April 4 election of Avenue 29 / Avenue 66 majorities of voters could culminate the bulk the government shutdown of all major projects on Manhattan's Westside – the means for those planned commuters to conduct their business.
The deteriorating job morale of union officials forbids opposition legislators from flying out on the New York state expressway bus.
The leaders of the unions need to go to law school, however. to denounce the proposal, a pattern well-held.
After a task force being drawn up by last year's Department of Transportation, the new policy would by train drivers and passengers carry the date of the start, the last intervening time of federal service, implemented in 1948.
That has been the policy for 65 years. The vast majority of these riders will come from the demanding sector. " idea is to pass for everyone, we pass for everyone," the governor said.
Eldermen have been allowed to have up to six seats on their all-board railings at the top, compared to non-disabled strollers and passengers. Some buses are reserved for seats of nearly the entire volume. - an arrangement that is familiar to a middle-management class set at the door during peak hours in the Manhattan region.
Commissioned officials, for signing familiar with the idea, will not be required to raise the fares – they will fly to work when new situations arise. Disabled workers and disabled passengers will also have to fly to work. Some assemblymen have spurned calls for state landowner leaders Joe Slinghaus and Bill Renfryant to build a four-speed train. But Cuomo could jump in with the proposal.
At least one¡—endoftext—¿

*Figure 18.* Sample Generation from CANDI at NFE=32 with temperature $\tau = .85$. At this NFE, CANDI generated samples with an average generative perplexity of $50.58$ and an entropy of $5.26$.

¡—endoftext—¿ "He has been enveloped in disarray[s] and is now very anxious to review his findings and to go beyond that," said Comey on the Tuesday report.

The FBI has said Nee's email exchange with O'Keefe created that it "a signifier" Hillary Clinton and her transition team's emails are "stolen" and "a sign of tampering" in the election. During the operation, the FBI has described Clinton's top aides as "a prime suspect" in the wake of "the troubling internal gaff" about Abedin's memo Comey's office would say was meant to "knock up" Comey's emails.

In an interview with Red Radar in Washington last week Tuesday, BuzzFeed reported that the FBI kept Clinton's private communications firmly sealed, allowing her to release secret-in-house information and unofficial communications of the Clinton transition team.

Former FBI Director James Comey appears to be the main antagonist in this election — identified as a "moderate" and behind Trump -, especially as Trump warned the FBI wouldn't abandon the Democrats and unite with them.

He said she had been "highly cooperative" and has denied waging any other effort to confront the president over his email server practices.

Mason asked for the emails to go "closed-on" when he attended a congressional probe of Infaroagate, Comey told PBS-Hour.

"As it led to evidence of fraud and in connection to O, who was the head of it,, asked to be cleared to face charges in public view, as is overreach of the Justice Department here," Comey said.

"It was impossible for any FBI agent to see the evidence," lawyer Susan C. Murphy, who worked for Clinton with Lynch, said on Monday.

"In searching for the truth and intelligence equipment that had closed the door on Clinton and using it to discredit her," she said, the FBI opened its investigation "becoming [itself] a temporary exercise for citizens to demonstrate how to end the moves of which Justice has not been taken hold."

Murdrick report also found connections between O and five other Clinton aides.

The report by the House and Senate Judiciary Committee investigating the Clinton "genocide" that the recipient of the attorney general and former Florida Gov. Scott Bush (R-Fla.) "was a letter designed to be seen as a resignation from the presidency he was yearning for rather than a distraction to the investigation."

Three Republicans said while they proceed committee hearing is expected to begin Wednesday. Democratic Vice President Joe Biden and John Kerry both sent the letter to the Clinton campaign, saying it has proven there were leaks and "likely ways in which the committee, above confidentiality, could have made the decision."

"In itself, Mueller's explanation is disappointing," the letter said "because in addition to the harm it would cause Sen. John McCain (Arizona) with his statement and decision, they have to intervene within the field within the field aimed to further harm those already in the field. They's no doubt sending a report to indicate that he has been wrong-footed. . . .

Even the Clinton state was enraged by the tone of the investigation when Clinton revealed that the cell phones seized she shared were leaked to the FBI.

"Non-confidential information is or should be public," the former president's interim said as Fox 24 reported on the alleged cover-up of the phone calls.

Though Mueller is legally barred from attending committee meetings, Clinton, the whose wife whom he met at the Palm meeting during November, reportedly stripped him of time in the day to testify as an evidence witness of "related sex crimes" and insisted that he and Clinton "follow directions" on a e-mail server.

Mueller's former House Intelligence Committee director, James Comey, faced criminal charges last year believing that Mueller was aware of a special interest between Trump's presidential campaign and and potentially his spouse. But the allegation refuted by Trump made in September of 2016 that Trump was "appealingly taking advantage of Secretary Clinton."

Attorney Loretta Lynch slammed Comey first, saying the investigation into the emails was "unprecedented and official evidence", and the letter "provided the promotional materials for further success in the investigation at this point."

Murdrick then recounted his experience, a "special agent and director of the FBI" watching and told Comey they "don't know."

Come, then, Comey.

"His team talked to me about those documents and I wanted to take off on¡—endoftext—¿

*Figure 19.* Sample Generation from CANDI at NFE=64 with temperature $\tau = .85$. At this NFE, CANDI generated samples with an average generative perplexity of $43.87$ and an entropy of $5.24$.

¡—endoftext—¿ third thing, "lean" is the right place to start.

I never had the chance to do that. I've tried to do it hard. It's not an attempt to do the right thing. So the end result is the job the people who are being paid and the ones who are turning out to be worth it. And that might not always be a shift-sinker coming in on the weekend.

The thing you need to work with is get the most up-to-date people. If you look at them with heart, intelligence, and re-witness them for a while, then in the end, you're the ones that've done well, and I'm also going to connect with them and give back to a lot of great people, which will be awesome, and that's all part of my journey of self-optimism.

So what do you want the corporate tech community to do for your hiring, and what do you have for that?

First, I don't want to have a telco job. I like to take two-to-one jobs in the sense that there's somebody out there pushing something that's really happening. You should get a good feeling it's almost over, which is when they announce publicly, "Now our shift-sinkers have to be our last ones." But I have to give them the relief that if it never DID lead me up to the end of the hire, then I might dismiss the whole idea. If I didn't leave the door open, I would come up with ideas, to get better feedback they would provide it, put shift-sinkers on board, put telco on board. All wrapped up, these guys said "we won't leave the board open for you.

I would like need to ask. What about you when I say as a boss, if you don't think it's like, through your hiring a tech worker? What's SO important

It's great. I'm excited to be joining a company leader. But in an actual HR department. You know how that works? Let's say, I have spent over+ years having to do 1,000+ of them. All 500 new employees. That would never be one employee in any organization. I know how HR people keep going out, interest you, they don't have to manage their own resources, they'll need someone at every time, or they'll move the resources quickly to what we need. But the private and the private sector where I have always worked from are actually incredibly strong. That's I have made quite some tactical changes during my years of an organization and past B.A. These things happen as we improve front-line hierarchy, door-to-door with a CEO that's committed to our company's.

So they tell me, "As we build this, we won't have any single thing that really makes that organization go nowhere, because it won't happen until we have company under-performing."

Yes, you are mentioning the fact that one of our top development officers right now is responsible for central management, so that would be useful for your position.

Absolutely true. Because if I did central management what happened, you'd be yelling at outsourcing scope-wise and we just ran as well as we were doing things. I can always talk about strategy. It's about making sure you know there's a responsibility of executing what you're going to do at any point in the proper context. And that is, when you think some quality products where we have this first and that experienced team I build around them, there's just too much responsibility. That's really not all about what I do. It's more about getting things done than just building up our operation to handle stuff and doing you better as you do it.

Let's need more responsibility, let's need even more patience!

If there are people who are struggling to get their jobs done because of Donald Trump's massive comments, then they are being able to help?

Because the person disappointing Donald Trump personally is that his comments about firing hit us all the hard nine times. That doesn't matter. The one thing I don't care about is that we've all lived pretty hard lives to get the gig. They can't save more time! It's not something that mattered to me. So I agree with that, to work out who I am doing.

The algorithm is very important. Previously, I had never been around to software engineers,¡—endoftext—¿

*Figure 20.* Sample Generation from CANDI at NFE=128 with temperature $\tau = .875$. At this NFE, CANDI generated samples with an average generative perplexity of $46.470$ and an entropy of $5.27$.

¡—endoftext—¿ for years and put innocent people in peril."

The case comes seven days after the indictment was held – a month too late - the first time bureau had closed a single gun-violence case in Ohio since 1998, a 20-year-long span started by Mark Millan, who recorded 14 attacks on Republicans.

The Justice Department earlier announced Thursday's charges and said one of its agents contacted Millan in 1998 when he was arrested in Falmouth.

The Justice Department did not respond to a phone call from Akron' State Attorney Troy Cummings, a spokesman in Washington.

Although Millan's actions were public knowledge and there is no criminal prosecution, the source said, it was rare the case was closed, Cummings said.

A representative in Washington, David Ruiz, with the FBI declined to requests for comment.

"Millan admitted that he had 'never been in a targeted situation,' he told reporters, speaking about the 1998 incident," Ruiz said. "He does not wish to give any details at this time."

Millan was not sought by the Justice Department, who said Millan has made statements on gun control. Moser said the FBI conceded "on observance of the investigation," his determination to prove Millan's acted unlawfully.

"The FBI was able to highlight significant facts and contribute to the investigation," Moser said.

In a statement to the Times, Issa said Millan's attacks against Terry McAuliffe in 1998 and "his actions were the actions of a mentally disturbed man and was a highly objective matter to be considered." He said they will bring "public safety into contact with us."

The Justice Department is not responding to a request from Akron for comment Thursday.

The Justice Department said it has asked the office Thursday to determine if there were any complaints, Moser said. Millan's lawyers were opening up their databases and culled a target of federal overreach, the bureau argued, and Issa and the office said they had taken no action that amounted to federal confiscation.¡—endoftext—¿During the weekend, the neighbor wrote that police believed for years in families not burning down houses to protect their children, according to her mother.

Police were called for fear of burning down homes and to burn the woman's home, said her mother, Jessica Medullo, the daughter of Adrianullo.

The neighbor took the family to other lives, where she has several grandchildren. Her kids were videotaped for fear of burning snow and then burning their houses, as she did on her own, Jessica said.

Jessica then told the paper she has tried to keep her elder daughters from ever becoming adults. She said that the children burned right up alive.

The woman still wanted all her children to die, "because they are the heart of the house," Jessica said.

Medullo told the paper she "was a child that my wife killed."

Medullo said that once the police arrived "she was willing to run over and kill the kids because they all were going home."

"That was amazing," the young woman wrote, Facebook page comments replete "risk-a-fire times."

Medullo is in the custody of children she adopted as a young girl with the help of her sister, she said in an email.

Yale¡—endoftext—¿San Diego, Arizona - This is actually what's great about the job of crusader.

Elizabeth Hastings was 93. Aug. 22. She worked at an apartment when her neighbors woke up, and they were called home by their owner, screaming as a "cheap." Her neighbors thought the woman had to have died, or she was unable to pick up food. When the only person called for medical help, she pushed open the door and opened a bookcase. Every time they mentioned how hot the room was, they used a pink bat because the rain started to fall, and then she shook her head and said, "Sorry for that bat!" We retired like other children and finally had the time to go doing a water down walks. The bat did not look spindly; the dog did not be dark and tough like dogs would have them; and had open eyes.¡—endoftext—¿O nacent Caucasians Americans were very resistant to mild experimenters hellfire. "We didn't know what would happen in our apartments in the Eighties. - Mrs. Dabaws

www.twitter.com/PlugInDangler pic.twitter.com/SufferinDin7D - U.Saver (@Canouse_Saver) June 19, 2016

She felt the loss of all the people she worked for, and when she came to defend her apartment with a fire, she¡—endoftext—¿

*Figure 21.* Sample Generation from CANDI at NFE=256 with temperature $\tau = .875$. At this NFE, CANDI generated samples with an average generative perplexity of 36.89 and an entropy of 5.21.

¡—endoftext—¿, was sending him a letter. The New York Post has also identified several of the people responsible for the statement.

As to the allegations, Ernst said that it hasn't been as formal allegations as Thapet's allegations were made. "There are tough questions about this, but we're reluctant to comment on the story. We don't know yet whether the people have fabricated the allegations. He got into a situation by trying to dress up'.

Though the media has investigated the case, and viewed a subsequent commentator on the soccer conference, David Pearlman, as intrigued, it would be far too late to agree to any fresh statements at the end of the window.

In June, Logue, having been fired by Treadwell, is set to join the club as part of a board of the league's first businessmen. Francis Marin, the winner of all 41 posts, will face felony charges in the case.

The club appears to be under attack for a new business — P.J. — after former CLARK Alnieval, the club's chief operating officer, was moved to New York City. Real estate chief, Mohamed Khan, struck a deal in Brooklyn, where he has been sitting, along with their executive chairman, Bruce Pearson. FC's Michael Fidez might be a chief executive with property development, but he says most of the club's supporters feel dismayed at the prospect.

"We came out with us remember Ali's passing and mourn Ali. We felt heartbroken at his death," said Wahida, of the irony of Jordan refusing to resign at that time. "The league shouldn't be too quick to let us post a full salary."

Membership has still managed to grow, with close to $1m, partly, thanks to their own troubled history, especially on the notorious firefighting raids of New Jersey.

It has caused international embarrassment throughout the club itself, which has also been focused internationally. FC's Holli Broadbent resigned in the spring, revealing that he ended up settling a bribery investigation by a jury that left the lower chamber in disarray. The men, well represented by the Justice Department's graft agency indicted a former trainer who had proven himself into foul play.

Brendan Frankel, the club's executive, wrote to President Barack Obama, who urged that moving outside the U.S. was "a political stunt". Frankel was impressed with Frankfurt. He said the move "will really be a shame on Ali and his family."¡—endoftext—¿Will England dug my teeth during the course of my career?

James Dobson, who won the Fulham Steve Sanders Award in last summer's FA Cup, had been linked with a Leeds job as the senior defender's assistant, having a partnership with Jose Mourinho.

But with Fulham banned his American rival from football, he was put in the dark, saying a split had been plan for Fulham, Leeds, Leeds and Exeter to follow. He didn't appear in England's final club the World Cup following this season's Leicester move.

You are over on the final day of your career.

There have been several other clubs speculating about the extent of my current role, and those are clubs I've never mentioned in the media. However, I spoke to Michael Jegren about what he thought about dressing room, and I spoke of my role at Melwood Fix It, Anger and Hope and Melwood's time at Footballing Club England. Last summer, there were some reporting me as having speculative reports of Andy Carroll's departure, and information will only trickle down to the club next season.

Now at Melwood, there has been not to me any cause for disappointment. Nathan Brekner is of relative stability, as he was brought back to north London several years ago, and I understand that there were a few occasions when I followed the same logic. Those clubs that are trying to go back on a steady basis have managed to get a chunk of money from some of their bigger American rivals.

I mean it would be a waste of time for the club to realize that a billion dollar team in the US market would not be available, and the bulk of the American team would have been picked up from the Senza Boys or Foxx, which are owned by some regions and would mean the team would not be available after about 15 million budget.

Do you feel that you are objecting to the clubs who wore gloves over the sky, including Avion, Mayweather and the Field Association for Sports, are you still conciliatory or wary?

It is especially instructive to understand basic rules that exist in football clubs and with other clubs, irrespective of what they¡—endoftext—¿

*Figure 22.* Sample Generation from CANDI at NFE=512 with temperature $\tau = .875$. At this NFE, CANDI generated samples with an average generative perplexity of 42.29 and an entropy of 5.21.

