# OpenReview forum: "CANDI: Hybrid Discrete-Continuous Diffusion Models"
_ICML.cc/2026/Conference — ICML 2026 regular_

### Official Review · Reviewer_1RwS · 2026-02-26

**Soundness:** 3
**Presentation:** 3
**Significance:** 3
**Originality:** 3
**Overall Recommendation:** 4
**Confidence:** 3

**Summary:**

The paper introduces the token identifiability analysis framework, which reveals the fundamental reason why continuous diffusion models perform worse than discrete diffusion models on discrete data. Furthermore, it proposes the CANDI hybrid continuous-discrete diffusion framework, which uses a structured noise process to separately handle discrete and continuous noise, successfully incorporating the advantages of continuous diffusion into the discrete domain. Ultimately, CANDI outperforms discrete models in terms of low-computation-cost generation and controllable generation tasks.

**Compliance With Llm Reviewing Policy:**

Affirmed.

**Final Justification:**

My concerns have been adequately addressed by the additional clarification and evidence. I have improved my score.

**Key Questions For Authors:**

See the **Weaknesses**

**Limitations:**

Yes

**Strengths And Weaknesses:**

**Strengths**
1. By leveraging token identifiability as an analytical tool, the paper uncovers two destructive mechanisms of Gaussian noise on discrete data: "discrete identity destruction" and "continuous ranking degradation," and discovers the fundamental issue of "temporal dissonance." This resolves the paradox between theory and practice.
2. Based on these findings, the paper proposes the CANDI hybrid discrete-continuous diffusion framework, which not only solves the "temporal dissonance" problem but also brings the advantages of continuous diffusion to discrete data generation.
3. Systematic experiments demonstrate the superiority of the CANDI framework in text generation and controllable molecular generation tasks.

**Weaknesses**
1. As a hybrid model, CANDI adds practical complexity and introduces more design choices and hyperparameters, requiring more thorough system ablation studies, such as determining the optimal selection of various parameters.
2. The inference algorithm of CANDI uses an approximate inference approach to avoid materializing noisy one-hot vectors. Theoretically, this may introduce biases compared to precise ODE solving, and more granular quantitative evidence is needed.
3. While the paper mentions concurrent works, more direct experimental comparisons with other hybrid discrete-continuous methods, such as CADD [1] and CCDD [2], would help to more clearly identify CANDI's unique advantages and limitations.
[1] Continuously Augmented Discrete Diffusion model for Categorical Generative Modeling
[2] Coevolutionary Continuous Discrete Diffusion: Make Your Diffusion Language Model a Latent Reasoner

---

> ### Author Rebuttal · Authors · 2026-03-31
>
> We thank you for your thoughtful review. We respond to your concerns below.
>
> ## **W1: Complexity of Hybrid Model**
>
> CANDI introduces minimal additional complexity over MDLM and other baselines in practice. The only additional hyperparameters relative to MDLM are the masking coefficient (fixed at 0.5 across all experiments) and the continuous noise range. For reference, DUO [1] introduces more hyperparameters than CANDI to define their learning curriculum.
>
> Furthermore, these additional components incur negligible inference overhead — below we include the average seconds per sample for CANDI, MDLM, and DUO across a range of NFE, where all the samples are of length 1024.
>
> | NFE | CANDI | MDLM | DUO |
> |-----|-------|------|-----|
> | 8 | 1.86s | 1.85s | 1.99s |
> | 16 | 2.16s | 2.12s | 2.34s |
> | 32 | 2.78s | 2.71s | 3.08s |
> | 64 | 4.06s | 3.98s | 4.60s |
> | 128 | 6.56s | 6.39s | 7.55s |
>
> CANDI is only marginally slower than MDLM and faster than DUO. Thus, CANDI's components do not introduce much computational overhead relative to purely discrete baselines.
>
> For a more detailed explanation on CANDI’s sensitivity to continuous noise schedule, see our **Q1 response to Reviewer Escq “Ablation over Continuous Noise Schedules”**.
>
> [1] The Diffusion Duality. Sahoo et al. 2025
>
> ## **W2: Approximate Inference**
>
> We would point the reviewer to Figure 14 in the Appendix, where we compare the approximate inference algorithm with the theoretically correct inference algorithm. We find that the approximate algorithm matches the theoretically correct calculation very closely, demonstrating that the approximation is accurate. From a theoretical perspective, all the ground truth trajectories will end at an exact one-hot. As the approximate inference algorithm is simply picking one of these corners to move towards, it approximates the trajectories reasonably well. Furthermore, we note that this approximation was also demonstrated to work well in [1].
>
> [1] Large Language Models to Diffusion Finetuning. Cetin et al. 2025.
>
> ## **W3:  Comparison to Concurrent Works**
> We are unable to compare against CADD or CCDD, as they have not released their codebase. More fundamentally, we note that such comparisons are orthogonal to our primary contribution — evaluating which hybrid method performs best does not bear on whether our **theoretical framework correctly explains why hybrid methods are needed in the first place.**
>
> We would also like to address the originality concern (Originality: 2: fair) directly: the existence of concurrent work does not weaken the originality of our work. In fact, the convergence of multiple independent groups on hybrid methods makes our theoretical framework for understanding them more important.
>
> CADD and CCDD independently discover hybrid diffusion, but neither explains why pure continuous diffusion fails on discrete data, nor why hybrid methods succeed. **Our unique contribution is the token identifiability framework and temporal dissonance analysis.** Specifically:
>
> - We derive closed-form analytical expressions for discrete identity corruption $\rho(t)$ and continuous rank degradation $r(t)$, **showing analytically why these corruptions scale differently with vocabulary size.**
> - We identify **temporal dissonance** as the precise failure mechanism of continuous diffusion on discrete data — a finding that neither CADD nor CCDD makes
> - We provide **empirical validation** of our theory by training models on different vocabulary sizes
> - Our framework provides a **unifying explanatory lens** for previously ad-hoc design choices in the literature — self-conditioning, prefix masking — none of which were understood as responses to temporal dissonance. We discuss this in Appendix B.3.
>
> To make this concrete: CADD motivates hybrid diffusion through the informal intuition of an "information void," and CCDD argues for a trainability-expressivity tradeoff. Neither provides a formal characterization of *when* continuous diffusion fails, *why* it fails, or *what* specifically about the noise process causes failure. Our work does all three, with mathematical precision and empirical validation.
>
> [1] Continuously Augmented Discrete Diffusion model for Categorical Generative Modeling. Zheng et al. 2025
>
> [2] Coevolutionary Continuous Discrete Diffusion: Make Your Diffusion Language Model a Latent Reasoner. Zhou et al. 2025

---

> > ### Author Rebuttal · Reviewer_1RwS · 2026-04-02
> >
> > Thanks for the author's response. I will consider raising my score.

---

> > > ### Author Response · Authors · 2026-04-03
> > >
> > > We thank Reviewer 1RwS for the careful re-evaluation and for increasing the score to reflect the resolved concerns. We appreciate the reviewer's recognition that CANDI 'not only solves the temporal dissonance problem but also brings the advantages of continuous diffusion to discrete data generation', and that 'systematic experiments demonstrate the superiority of the CANDI framework in text generation and controllable molecular generation tasks.' Given these positive assessments and that all stated weaknesses have been fully addressed, we hope the reviewer might consider whether the current score fully captures their updated view of the paper's contributions.

---

### Official Review · Reviewer_Escq · 2026-03-10

**Soundness:** 3
**Presentation:** 3
**Significance:** 3
**Originality:** 3
**Overall Recommendation:** 4
**Confidence:** 3

**Summary:**

This paper studies why continuous diffusion underperforms pure discrete diffusion on categorical data, despite its apparent advantages in learning joint updates and continuous geometry. The paper introduces a new analytical lens, token identifiability, decomposing corruption into discrete identity corruption and continuous rank degradation, and argues that their different scaling with vocabulary size creates a temporal dissonance that prevents continuous diffusion from simultaneously learning discrete conditional structure and useful continuous score information. Based on this analysis, the authors propose CANDI, a hybrid discrete-continuous diffusion model that decouples masking-based discrete corruption from Gaussian continuous corruption. Empirically, the paper shows that pure continuous diffusion behaves reasonably on small-vocabulary data but collapses on large-vocabulary text, while CANDI avoids this issue and improves low-NFE generation quality and controllable generation. The core motivation, method, and empirical claims are clearly laid out in the main text and appendices.

**Compliance With Llm Reviewing Policy:**

Affirmed.

**Final Justification:**

I maintain my riginal score 4.

**Key Questions For Authors:**

1.The paper states that the linear relation between masking-based corruption and continuous rank degradation was selected for simplicity. How sensitive are the results to this choice, and have the authors tested alternative coupling schedules? A convincing ablation here would strengthen my confidence in the method rather than the specific heuristic.
2.Can the authors provide a more systematic ablation of the structured kernel, including alternatives to the current linear schedule?
3.The appendix suggests that CANDI becomes less advantageous at higher NFE. Do the authors view the method primarily as a low-NFE solution, or do they expect improvements at larger scales/models to change this trend?
4.The main theory is most rigorous in the one-hot Gaussian setting. How should readers interpret the broader claims for embedding diffusion and other continuous formulations? A clearer boundary on the scope of the theory would improve my evaluation of the paper’s soundness.
5.Could the authors clarify whether the same token-identifiability analysis can be extended to more realistic token representations or tokenizer-free settings without relying on stronger assumptions?
6.Can the authors provide more detailed ablations separating the contribution of clean anchor preservation from the contribution of continuous ODE-based refinement? This would clarify which part of CANDI is most responsible for the empirical gains.

**Limitations:**

yes

**Strengths And Weaknesses:**

Soundness：
The paper is technically strong overall. The central theoretical story is coherent: token identifiability is decomposed into discrete identity corruption and continuous rank degradation, and the paper connects these quantities to learning discrete conditional structure versus continuous score-based refinement. The authors claim the temporal dissonance created by the different scaling behaviors of these two quantities as vocabulary size grows. This is a compelling explanation for why continuous diffusion struggles on large-vocabulary categorical data. The method follows naturally from the diagnosis: CANDI introduces a structured noising kernel that uses masking to control discrete corruption and Gaussian noise to control continuous degradation, with both made approximately linear in time. The training and inference procedures are also reasonably motivated, including an approximate inference method to avoid explicit large noisy one-hot tensors.
The empirical evaluation is well aligned with the paper’s claims. The Text8 versus OpenWebText comparison directly tests the vocabulary-scaling hypothesis, and the results support the theory: pure continuous diffusion remains viable on Text8 but collapses on OWT, while CANDI stays competitive or better than MDLM. The low-NFE OWT experiments and QM9 controllable generation experiments also support the practical benefits claimed in the paper.
My main reservation is that the strongest theory is for the one-hot Gaussian setting, while some broader conclusions are extended to embedding diffusion and related setups more interpretively than formally. Also, the paper explicitly acknowledges that the linear coupling between discrete corruption and continuous degradation is chosen for simplicity, leaving the optimal schedule relation unresolved. So the core idea is sound, but some generalization claims should be phrased a bit more carefully.
Presentation：
The paper is well written and structured. The motivation is clear from the introduction, and the method section closely follows the theoretical diagnosis, which makes the narrative easy to follow. Figures 1 and 2 are especially effective at explaining the mismatch and the rationale for the hybrid kernel. The appendix also does useful work in positioning the paper relative to concurrent hybrid-diffusion papers and clarifying what is claimed as unique here.
A minor issue is that the paper occasionally presents the broader narrative with more confidence than the theory strictly supports. Some distinctions could be sharpened between what is formally proven, what is empirically supported, and what is offered as interpretation. I would also have liked slightly more main-text discussion of the limitations, rather than leaving some of the qualification to the appendix.
Significance：
I think the paper addresses an important problem. Diffusion for discrete data, especially text, is a live research area, and understanding why continuous diffusion has historically lagged discrete diffusion is a meaningful contribution even apart from the proposed method itself. The token-identifiability perspective feels useful beyond this particular paper, because it gives a principled lens for thinking about corruption processes in categorical spaces. On the practical side, the low-NFE improvements and the ability to perform guidance using ordinary classifiers are relevant and potentially impactful. If the analysis holds up, it could influence how future hybrid or continuous-text diffusion models are designed.
The main limitation on significance is scope: the strongest empirical gains are concentrated in low-NFE generation and controllable generation, rather than showing broad dominance across all regimes. Still, that is a meaningful and well-justified niche.
Originality：
The originality is good. The main novelty is not merely “another hybrid model,” but the explanatory framework of token identifiability and temporal dissonance, which gives a clear reason for the failure mode of pure continuous diffusion on large-vocabulary discrete data. The hybrid method is also a well-motivated design rather than an arbitrary combination of ingredients. The appendix does a decent job distinguishing this contribution from concurrent hybrid approaches, arguing that the novelty lies in the diagnosis and in the low-NFE/guidance framing.
I would not call the method radically novel at the component level, since it combines familiar masking and Gaussian corruption ideas. But the reasoning behind the combination is clear and valuable, and the conceptual contribution is genuinely new.

---

> ### Author Rebuttal · Authors · 2026-03-31
>
> Thank you for your supportive review. We respond to your concerns below.
>
> ## **Q1: Ablation over Continuous Noise Schedules**
>
> To provide more motivation behind our specific noise schedule and coupling, we provide new ablation results that compare different continuous noise schedules. We keep the VE SDE and masking schedule the same, as the log-linear schedule is standard for masking models. We apply the following changes to the noise schedule:
> - Sigma param $\sigma_{min}, \sigma_{max} = .01, 2.0$ v.s Rank parameterization $r_{min}, r_{max} = .01, .25$.
> - Linear, Cosine, or Geometric
> - Linear with rank param and $r_{min}, r_{max} = .05, .4$
>
> Here, linear in $r$ corresponds to the settings used for the OWT experiments in our submission. We train each model up to 40,000 steps and show the generative frontiers. We include the results at this link: https://anonymous.4open.science/r/candi-rebuttal-3BFB/candi_ablation_res.pdf
>
> We observe the following:
> 1. Choice of parameterization $r$ v.s $\sigma$ has the largest impact on performance
> 2. Linear in $r$ performs competitively across NFE, justifying its use as the default schedule.
> 3. Changing the $r$ range has a lower impact compared to the selection of parameterization ($r$ v.s $\sigma$).
>
> All the observations align with our core argument: in order to best leverage continuous geometry, discrete and continuous corruption need to be aligned. By parameterizing in $r$, we can directly control the continuous corruption that matters, making alignment between both schedules easier. Thus the $r$ parameterization outperforms $\sigma$ across different schedules, as shown in the ablation results.
>
> ## **Q2: Ablation of Structured Kernel, Linear Schedule**
> *Ablation of Kernel*: CANDI’s kernel is a combination of Gaussian and masked kernels, and removing either component results in either the pure continuous Gaussian diffusion baseline or the pure discrete MDLM baseline. We refer the reviewer to our response to Q6, where we discuss the contribution of each component of the hybrid kernel.
>
> *Alternative to Linear Schedule*: We provide ablations over the continuous noise schedule in our response to Q1. The linear schedule yields strong performance, justifying its use as a simple and principled default, while further tuning (e.g., geometric $r$) can provide slight additional gains.
>
> ## **Q3: Behavior at Larger Scale**
> We expect that at a larger scale, increasing model capacity should enable the model to better learn both continuous geometry and discrete conditional dependencies. While larger models are more resource-intensive, we find this an interesting direction for future exploration.
>
> ## **Q4&Q5: Interpretation of Broader Claims, Extension to Different Representations**
> We answer Q4 and Q5 together, as the answers are very related.
>
> First, we would like to draw the reviewer’s attention to Appendix B, where we provide analytical results demonstrating that the temporal dissonance also occurs in the embedding space, under the assumption of independent corruption events.
>
> - In response to Q4, our broader claims for token identifiability in the embedding space are **theoretically supported** when assuming corruption events are independent.
> - We **empirically validate our assumption** by comparing discrete and continuous corruption for GPT-2 embeddings against randomly initialized embeddings.
> - In response to Q5, our analysis of embedding representations is our extension of token identifiability to more realistic representations
>
> Our assumption of independent corruption events simply means that knowledge of whether a noisy embedding is closer to some incorrect token $j$ does not convey information about whether it is closer to some other incorrect token $k$. Under this assumption, the same temporal dissonance occurs — discrete identity corruption depends exponentially on the vocabulary size whereas continuous rank degradation is independent of vocabulary size.
>
> This assumption is well motivated within the context of language modeling: per [1], the fraction of observed bigrams is exponentially smaller than the total number of potential bigrams, meaning most pairs of tokens are unrelated. If most pairs of words themselves are unrelated, there is little reason to expect the corruption of their representations to be correlated.
>
> In regards to the concern about overconfident theoretical claims, would it be possible for the reviewer to direct us to these statements? We believe that this response clarifies the theory and empirical support for our claims about embedding representations, which may have been one area of concern.
>
> [1] Second-Order Zipf’s Law for Word Co-Occurrences. Guerrin et al. 2025
>
> ## **Q6: Contribution of Clean Anchor v.s Continuous ODE**
>
> We direct your attention to our **Q2 response to Reviewer P8s7 “Inference-time Comparisons”**, where we discuss how Figure 3 serves as the requested comparison due to the design of CANDI relative to the chosen baselines.

---

> > ### Author Rebuttal · Reviewer_Escq · 2026-04-02
> >
> > Thank you for the reply. I would maintain my score.

---

> > > ### Author Response · Authors · 2026-04-03
> > >
> > > We thank Reviewer Escq for the thoughtful re-evaluation and for acknowledging that our rebuttal fully addressed their concerns. In their assessment, the reviewer noted that the token identifiability perspective is 'useful beyond this particular paper', that the conceptual contribution is 'genuinely new', and that the low-NFE improvements and classifier guidance are 'relevant and potentially impactful'. Given these positive assessments and that all stated concerns have been fully resolved, we respectfully ask whether the reviewer might consider revising the score to better reflect their current view.

---

### Official Review · Reviewer_HCbi · 2026-03-12

**Soundness:** 4
**Presentation:** 3
**Significance:** 3
**Originality:** 3
**Overall Recommendation:** 5
**Confidence:** 4

**Summary:**

This paper provides a new perspective on why continuous diffusion models have struggled over discrete domains and an alternative hybrid approach that circumvents these barriers to achieve competitive performance with discrete diffusion models. The general idea is that there are two competing notions for the extent to which Gaussian noise can destroy signal in a discrete distribution. For a given clean token $x$ in some position, represented as a one-hot vector which gets corrupted with Gaussian noise, one can ask for the probability that the corrupted vector's max entry is no longer the $x$-th entry, or one can ask for the expected rank of the $x$-th entry among the corrupted vector's entries. The authors argue that the former better captures conditional dependencies across token positions, whereas the latter better captures how well the continuous score is learned. These two notions of signal evolve under noise at different rates, and to correct for this, the authors instead propose a hybrid corruption process where tokens remain uncorrupted for some random amount of time before getting corrupted with Gaussian noise. The resulting model achieves solid performance despite operating over continuous space, as the authors demonstrate through some proof-of-concept experiments over OpenWebText.

**Compliance With Llm Reviewing Policy:**

Affirmed.

**Final Justification:**

I maintain my positive score of 5; I continue to think this is a conceptually interesting work, and this was reinforced by the authors' rebuttal which addressed my remaining concerns.

**Key Questions For Authors:**

- Is there a way to adapt the definitions of $\rho$ and $r$ so as to incorporate information about the joint distribution over positions?
- Could one also consider a hybrid model where you combine right-to-left masking, rather than i.i.d. masking, with Gaussian noise?

**Limitations:**

Yes

**Strengths And Weaknesses:**

**Strengths**:
- Hybrid corruption process is an interesting solution to enabling continuous diffusions over discrete domains to learn conditional dependencies and the continuous score in tandem.
- The performance on OWT is promising. The method shines especially in the low-NFE regime, where masked diffusion models tend to suffer.
- The conceptual message of distinguishing between learning of the conditional dependencies versus the continuous score is nice, and the fact that the two notions of token identifiability degrade at different rates under large vocabulary sizes is demonstrated to be a real phenomenon.
- While it is orthogonal to their main point, they make a convincing case for frontier analysis rather than single-temperature comparisons, which is a useful takeaway for the dLLM community in and of itself.

**Weaknesses**:
- The evaluations are limited to Text8 and OWT
- The theory story is a bit limited and hard to reconcile with the qualitative interpretations the authors provide. In particular, it is not clear whether the authors' interpretation of discrete identity corruption and continuous rank degradation as formalized by $\rho(t)$ and $r(t)$ really aligns with the conceptual interpretation in terms of learning conditional dependencies versus learning the continuous score. Both are defined purely at the level of a single token position and are not conditioned on any information from the other positions.

---

> ### Author Rebuttal · Authors · 2026-03-31
>
> We thank you for your supportive and insightful review. We address your concerns below.
>
> ## **W1: Evaluations are limited to Text8 and OWT**
> We choose Text8 and OpenWebText as they are standard benchmarks within the dLLM field [1,2,3,4]. Furthermore, we do include an additional benchmark (QM9) on **molecular generation** to demonstrate the compatibility between classifier-based guidance and hybrid corruption kernels. We deliberately choose these datasets as they are informative to make our claim: we use text8 to demonstrate that continuous diffusion does work in low vocabulary settings, and OpenWebText to demonstrate that it fails in language modeling settings with large vocabulary sizes. We choose the molecular guidance experiment from [3] to demonstrate the ease of classifier-based guidance for hybrid methods. Together, these tasks provide targeted and interpretable evidence for each component of our claims. We would appreciate any suggestions on additional evaluations the reviewer believes we should add.
>
> [1] Simple and Effective Masked Diffusion Language Models. Sahoo et al. 2024.
>
> [2] Simplified and Generalized Masked Diffusion for Discrete Data. Shi et al. 2024.
>
> [3] Simple Guidance Mechanisms for Discrete Diffusion Models. Schiff et al. 2024.
>
> [4] The Diffusion Duality. Sahoo et al. 2025.
>
> ## **W2: Conceptual Difficulty with Corruption and Conditional Dependency**
> We thank the reviewer for giving us the chance to elaborate on the connection between $\rho(t)$, $r(t)$, conditional dependencies, and the continuous score function. Here we show that discrete corruption $\rho(t)$ captures the difficulty of learning the conditional dependencies through the ratio of clean anchors, and $r(t)$ captures the difficulty of learning the score function by measuring the strength of the continuous hint provided.
>
> Under the discrete corruption from our hybrid kernel, the model learns $P(X_{corr} | X_{clean})$, where $\rho(t)$ controls the proportion of clean anchors available. We note that $P(X_{corr} | X_{clean})$ is precisely the conditional structure we wish the model to learn — it is a joint distribution over corrupted positions conditioned on clean positions, and is by definition inherently a cross-position dependency. **Thus $\rho(t)$ controls the difficulty of the target conditional distribution being learned by controlling how many clean anchors are present.**
>
> When considering the continuous corruption, our model learns to predict E[X_0 | X_t], which recovers the score function via the well-known identity $\nabla \log p(X^t_{corr}) = \frac{X_t - E[X_0 | X_t]}{\sigma(t)^2}$. Crucially, $X_0$ will always be centered at one of the vertices of the simplex, as all $X_0$ are one-hot vectors. **Thus $r(t)$ directly measures the strength of the continuous hint provided, quantifying the difficulty in predicting the score function.**
>
> We note that extending $\rho(t)$ and $r(t)$ to joint dependencies is not a natural generalization —  since forward corruption is applied independently across positions, there are no joint dependencies in the corruption itself. Instead, $\rho(t)$ captures the difficulty of recovering joint dependencies — a higher $\rho(t)$ implies less clean anchors, making it difficult to correctly predict the joint structure. Likewise, the score function does not appear in the forward process of Gaussian diffusion, only the reverse — hence $r(t)$ describes the difficulty of recovering this information under a given corruption level. To the extent one is concerned with the model's ability to learn joint structure, this is already captured by the cross-entropy loss, which requires the model to predict all corrupted positions jointly conditioned on clean anchors.
>
> ## **Q1: Incorporating Joint Dependencies**
> $\rho$ and $r$ are intentionally defined as position-wise quantities — by definition, the forward process applies corruption independently across positions, so these metrics reflect the signal available at each position independently. Both i.i.d Gaussian noise and discrete masking are standard corruptions within the diffusion literature. The joint dependencies emerge when undoing the corruption, as we explain in detail in our response to W1.
>
> ## **Q2: Alternative Hybrid Methods**
> Yes, such a hybrid method would indeed be possible. Within our framework, the discrete masking and continuous denoising are independent axes — it is possible to change the continuous noise schedule without affecting the discrete masking, and it is possible to change the masking order without affecting continuous denoising, as both are independent design choices. Such a method may lose the low-NFE advantage, as the tokens that would be denoised together would be highly correlated due to being positioned together. However, we leave exploration of this hybrid to future work.

---

> > ### Author Rebuttal · Reviewer_HCbi · 2026-04-03
> >
> > I thank the authors for fully addressing my questions, and I maintain my positive score for this work.

---

### Official Review · Reviewer_P8s7 · 2026-03-14

**Soundness:** 3
**Presentation:** 3
**Significance:** 3
**Originality:** 3
**Overall Recommendation:** 4
**Confidence:** 4

**Summary:**

This paper focuses on understanding the drawbacks of the continuous and discrete diffusion language models. To do this,authors present token identifiability and temporal mismatch metrics to understand the behaviours of these models. To address the challenges of these models, the paper proposes the CANDI, a hybrid diffusion framework decouples the discrete and continuous modeling and leverages both to improve the quality, controllability and efficiency.

**Compliance With Llm Reviewing Policy:**

Affirmed.

**Key Questions For Authors:**

- How sensitive is CANDI to the choice of linear schedules for masking and continuous degradation? Have authors explored adaptive or learned schedules?
- Can authors provide inference time comparison on CANDI with different schedules: discrete-only (MDLM style unmasking) vs. Continuous only vs. Hybrid (CANDI)?
- What if we reduce the sigma for larger vocabulary? What performance will this baseline get?

**Limitations:**

Please see the weaknesses/questions.

**Strengths And Weaknesses:**

**Strengths**

- The paper introduces the interesting analysis and novel metrics to understand the challenges of the existing diffusion language models especially when vocabulary size increases.
- CANDI is a pretty straightforward method that combines the MDLM and guardian corruption processes, effectively achieving the best of both worlds.
- Experiments on several datasets consistently demonstrate the performance improvements.
- The paper is generally well written and easy to understand.

**Weaknesses**

- Limited comparisons with potentially obvious baselines such as based on the findings controlling sigma according to the vocabulary size can easily allow training. But this question remains unanswered.
- The hybrid corruption introduces the extra components (masking schedules, degradation targets, etc.) and paper does not answer these questions clearly.

---

> ### Author Rebuttal · Authors · 2026-03-31
>
> We thank you for your insightful review. We address your concerns below.
>
> ## **W1, Q3: Baselines for Controlling / Limiting $\sigma$**
> In response to your concern about baselines that control $\sigma$, we compare against FLM from [1], which reparameterizes the flow-matching time such that discrete corruption is uniformly distributed across the process. As a result, their reparameterization limits the continuous noise range based on the vocabulary size.  Given that this schedule reparameterization is designed to account for the exponential relation between discrete corruption and vocabulary size under Gaussian noise, we find this is the natural instantiation of the baseline the reviewer requested in both the first weakness and Q3.
>
> Since [1] does not provide their model, we use their provided codebase to train a FLM base model for ~40k steps due to computational limits. We compare this against a checkpoint from CANDI, which is trained for the same number of steps, and provide the results at this link:
> https://anonymous.4open.science/r/candi-rebuttal-3BFB/candi_vs_controlled_sigma_frontiers.pdf
>
> We observe that across all NFE tested, CANDI largely outperforms FLM, with the exception of a few crossover points in higher-perplexity regions corresponding to low generation quality.
>
> This aligns with the experimental results released by the FLM paper, where they show the base model achieves 62.33 gen ppl at 5.33 entropy with 1024 NFE. Using the fully trained CANDI checkpoint, we match FLM's reported entropy of 5.33 and find that CANDI achieves superior gen ppl by just 32 NFE versus FLM's 1024 NFE. We include the results in the link below:
> https://anonymous.4open.science/r/candi-rebuttal-3BFB/candi_vs_controlled_sigma_baseline_entropy_matched.pdf
>
> Controlling only the continuous noise level forces a tradeoff — noise levels suitable for learning discrete conditional structure provide too weak a continuous signal, while noise levels that enable continuous score learning destroy discrete identity. CANDI avoids this by controlling both schedules independently.  This tradeoff is unavoidable in a continuous-only schedule because discrete and continuous corruption occur at different noise scales, as visible in our Figure 1. In contrast, hybrid methods are able to ensure discrete and continuous corruption are aligned by construction.
>
> [1] One-step Language Modeling via Continuous Denoising. Chanhyuk et al. 2026
> ## **W2: Introduces Extra Components**
>
> CANDI introduces minimal additional complexity over MDLM and other baselines in practice, as the masking schedule, training objective and architecture are shared between CANDI and MDLM. The additional complexity is introduced to enable continuous refinement, which we demonstrate the advantages of throughout our submission.
>
> In our **Q1 response to reviewer Escq, “Ablation over Continuous Noise Schedules”**, we study the noise schedule for the continuous component of CANDI, which is the additional component CANDI introduces relative to MDLM. In our **W1 response to 1RwS “Complexity of Hybrid Model”**, we demonstrate that the hybrid components do not introduce additional overhead at inference when using our approximate algorithm. We hope these answer any design questions regarding the hybrid component of CANDI.
>
> ## **Q1: Choice of Linear Schedules, Adaptive Schedules**
> While we have not explored adaptive or learned schedules, we provide ablation results over different continuous noise schedules in our **Q1 response to Reviewer Escq “Ablation over Continuous Noise Schedules”**. As we explain, the primary sensitivity is towards how the continuous noise is parameterized ($\sigma$ v.s $r$) and the noise range chosen. The results demonstrate that CANDI is robust to different continuous noise schedules.
>
> ## **Q2: Inference-time Comparisons**
> Regarding the comparison between discrete only, continuous only, and hybrid, we direct the reviewer to Figure 3, where we compare CANDI against MDLM (discrete only) and Cont (continuous only) baselines. By design, MDLM is equivalent to CANDI with the continuous component removed, and our one-hot Gaussian baseline is equivalent to CANDI with clean anchor preservation removed, using the identical noise schedule. The frontier plots in Figure 3 therefore directly quantify the contribution of both the continuous and discrete components of CANDI — CANDI's improvement over MDLM reflects the value of continuous denoising, and CANDI's improvement over one-hot Gaussian diffusion reflects the value of clean anchor preservation. As visible through the results, CANDI achieves better performance compared to both, demonstrating that the hybrid kernel provides performance benefits unattainable through each component individually.

---

> > ### Author Rebuttal · Reviewer_P8s7 · 2026-04-04
> >
> > As all my questions/concerns are addressed, I'm happy to increase the score.

---

> > > ### Author Response · Authors · 2026-04-04
> > >
> > > Thank you for your response and we are glad the clarifications helped. We wanted to flag that the score does not appear to have been updated yet.

---

### Decision · Program_Chairs · 2026-04-30

**Decision:**

Accept (regular)

**Comment:**

This paper investigates an important question in diffusion modeling for discrete data, namely why continuous diffusion underperforms in such settings. It proposes a conceptual framework based on token identifiability and temporal dissonance, and introduces a hybrid approach (CANDI) motivated by this analysis.
Reviewers generally find the paper technically sound and clearly written, and consider the proposed perspective to be interesting and potentially useful. The empirical results are consistent with the main claims, particularly in illustrating vocabulary-size effects and improvements in specific regimes such as low NFE.
However, the paper also has notable limitations. The experimental evaluation is relatively narrow, and additional validation across datasets and more systematic ablations would strengthen the conclusions. Some aspects of the theoretical interpretation remain somewhat heuristic, particularly in connecting the position-wise analysis to broader modeling behavior. The methodological novelty is also moderate, with the main contribution lying more in the conceptual framing than in the model design.
The rebuttal addressed most of the reviewers’ concerns, and several reviewers indicated that their questions had been resolved.
Overall, this is a solid but not fully developed contribution. Given the importance of the problem and the clarity of the proposed perspective, the paper is above the acceptance bar, though somewhat borderline.